# Upconversion NIR-II fluorophores for mitochondria-targeted cancer imaging and photothermal therapy

Hui Zhou [1,2,7], Xiaodong Zeng [1,3,7], Anguo Li[1], Wenyi Zhou[1,3], Lin Tang[1], Wenbo Hu [4], Quli Fan [4], Xianli Meng[5], Hai Deng [6], Lian Duan[1], Yanqin Li[1], Zixin Deng[1], Xuechuan Hong [1,2✉] & Yuling Xiao [1,3✉]

NIR-II fluorophores have shown great promise for biomedical applications with superior in vivo optical properties. To date, few small-molecule NIR-II fluorophores have been discovered with donor-acceptor-donor (D-A-D) or symmetrical structures, and upconversion-mitochondria-targeted NIR-II dyes have not been reported. Herein, we report development of D-A type thiopyrylium-based NIR-II fluorophores with frequency upconversion luminescence (FUCL) at ~580 nm upon excitation at ~850 nm. H4-PEG-PT can not only quickly and effectively image mitochondria in live or fixed osteosarcoma cells with subcellular resolution at 1 nM, but also efficiently convert optical energy into heat, achieving mitochondria-targeted photothermal cancer therapy without ROS effects. H4-PEG-PT has been further evaluated in vivo and exhibited strong tumor uptake, specific NIR-II signals with high spatial and temporal resolution, and remarkable NIR-II image-guided photothermal therapy. This report presents the first D-A type thiopyrylium NIR-II theranostics for synchronous upconversion-mitochondria-targeted cell imaging, in vivo NIR-II osteosarcoma imaging and excellent photothermal efficiency.

[1] State Key Laboratory of Virology, Key Laboratory of Combinatorial Biosynthesis and Drug Discovery (MOE), Hubei Provincial Key Laboratory of Developmentally Originated Disease, Hubei Province Engineering and Technology Research Center for Fluorinated Pharmaceuticals, Wuhan University School of Pharmaceutical Sciences, Wuhan 430071, China. [2] College of Science, Innovation Center for Traditional Tibetan Medicine Modernization and Quality Control, Tibet University, Lhasa 850000, China. [3] Shenzhen Institute of Wuhan University, Shenzhen 518057, China. [4] Key Laboratory for Organic Electronics and Information Displays & Institute of Advanced Materials, Nanjing University of Posts and Telecommunications, Nanjing 210023, China. [5] Innovative Institute of Chinese Medicine and Pharmacy, Chengdu University of Traditional Chinese Medicine, Wenjiang, Chengdu, Sichuan 611137, China. [6] Department of Chemistry, University of Aberdeen, Aberdeen, UK. [7] These authors contributed equally: Hui Zhou, Xiaodong Zeng. ✉email: xhy78@whu.edu.cn; xiaoyl@whu.edu.cn

Fluorescence imaging in the second near-infrared window (NIR-II, 1000–1700 nm) has been widely used in biological and biomedical research, because the light in this wavelength region is capable of deep penetration, and cellular autofluorescence is minimal[1–17]. Our initial progress toward the first small-molecule organic NIR-II dye entailed the formation of the fluorophore CH1055 from benzo-bis(1,2,5-thiadiazole) (BBTD), which absorbed light at ~750 nm and fluoresced at ~1055 nm[18–20]. The LUMO of CH1055 showed a strong contribution at the electron-accepting moiety BBTD with a typical optical bandgap (Egap) smaller than 1.5 eV, which was beneficial for bathochromic shift and large Stokes shift. Despite a panel of NIR-II emissive contrasts have been explored, the majority of them had a BBTD core or were symmetrical fluorophores with a D–A–D structure (Fig. 1a)[21–29]. It is imperative to search for new types of small-molecule NIR-II probes with desirable chemical and physical properties, minimal cellular toxicity, and clinical translation ability (Fig. 1).

Among all the fluorescent dyes, frequency upconversion luminescence (FUCL) materials can convert low-energy excitation to high-energy emission with large anti-Stokes shift, limited auto-fluorescence from biological samples, high signal-to-noise ratio, and tunable excitation and emission[30–32]. Mitochondria are vital organelles responsible for the energy production and programed cell death such as apoptosis[33–35]. Despite the development of mitochondria-targeted therapeutics, imaging of mitochondria is also crucial for providing important information about disease presence, progression, and pathways for early disease diagnosis and treatment[36–39]. To date, limited examples of mitochondria-targeted FUCL organic dyes have been reported and used for deep-seated tumor treatment[40]. Furthermore, no research based on the FUCL and mitochondria-targeted NIR-II imaging technology has been developed for tumor imaging and antitumor therapy. Many small-molecule mitochondria-targeted agents have been developed, including alkyltriphenylphosphonium cations, rhodamine, and both natural and synthetic mitochondria-targeted peptides[38]. However, most currently available mitochondria-targeted fluorescent dyes emit only one color in the visible or NIR-I window. Their applications are somewhat limited due to poor photostability, small Stokes shifts,

or high-energy laser excitation. Therefore, the exploitation of new mitochondria-targeted NIR-II fluorophores with a more biocompatible upconversion strategy is greatly desired[41].

As useful and versatile heterocyclic luminescent compounds with a positive charge, pyrylium salts are widely utilized as photosensitizers, Q-switchers, and NIR dyes for proteins and oligosaccharide labeling[42,43]. Thiopyrylium is analogous to the pyrylium cation with the oxygen atom replaced by a sulfur atom. Notably, thiopyrylium salts display physical and chemical properties typical of aromatic compounds and are much less reactive than analogous pyrylium salts due to the lower electronegativity of the sulfur atom[44,45]. As is clearly seen in Supplementary Table 1, the Egap of thiopyrylium is much lower than that of pyrylium, suggesting that thiopyrylium salts may become novel NIR acceptors. In the current work, we utilized thiopyrylium heterocycle 2 as a building block to construct a series of novel red-shifted thiopyrylium dyes 3 with donor–acceptor (D–A) structures (Fig. 1b). The electronic structure calculations and experimental evaluation of the optical properties of 3a–3k and H4 were carried out to elucidate the structure–property relationship. Furthermore, the synthesized NIR-II fluorophores 3k-PEG, H4-PEG, and H4-PEG-PT were highly resistant to photobleaching and well retained within the mitochondria even after osteosarcoma cell fixation and permeabilization at 1 nM based on FUCL strategy with a short-wavelength luminescence at 580 nm upon excitation at ~850 nm, which is among the most favorable characteristics of fluorescent dyes. Moreover, H4-PEG-PT exhibited potent photothermal therapy effect toward osteosarcoma both in vitro and in vivo. To the best of our knowledge, this report presents the first D–A type thiopyrylium NIR-II theranostics for synchronous FUCL mitochondria-targeted cell imaging with subcellular resolution, deeper tissue NIR-II osteosarcoma imaging in vivo and excellent photothermal efficiency.

## Results

**Synthesis and rational design.** We first designed and synthesized 3a–3f ($R_2$ = H) to confirm our hypothesis that these dyes followed the D–A rather than D–A–D structure[46,47]. The chemical structures of 3 are shown in Fig. 1b. The synthetic route for 3 is shown in Fig. 2a. The key intermediate thiopyrylium

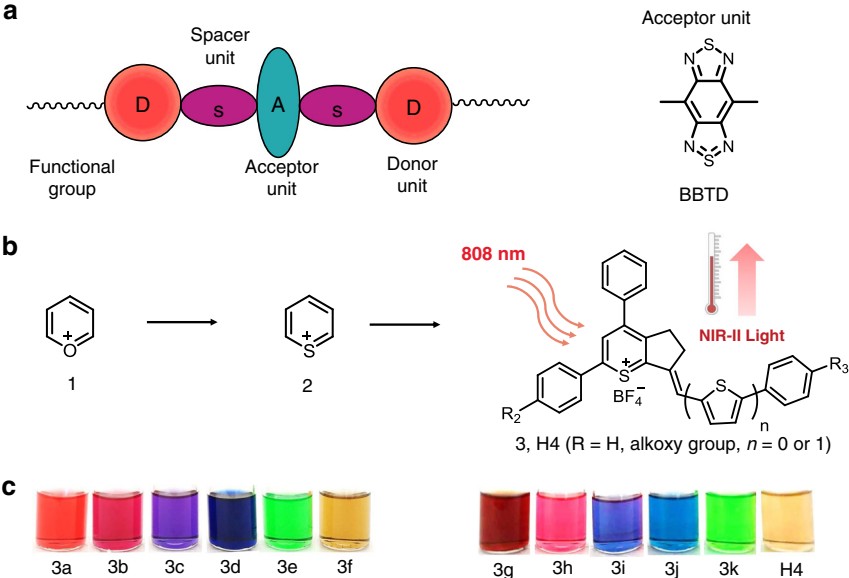

**Fig. 1 Structural design and bright-filed images of 3a–3k, H4. a** Traditional NIR-II small-molecule dyes based on the BBTD core with D–A–D or symmetrical structures. **b** A new strategy for the construction of D–A type thiopyrylium mitochondria-targeted NIR-II dyes with FUCL and photothermal properties, $n = 0$, 1. **c** Bright field images of a series of new thiopyrylium-based probes 3a–3k and H4 in dichloromethane.

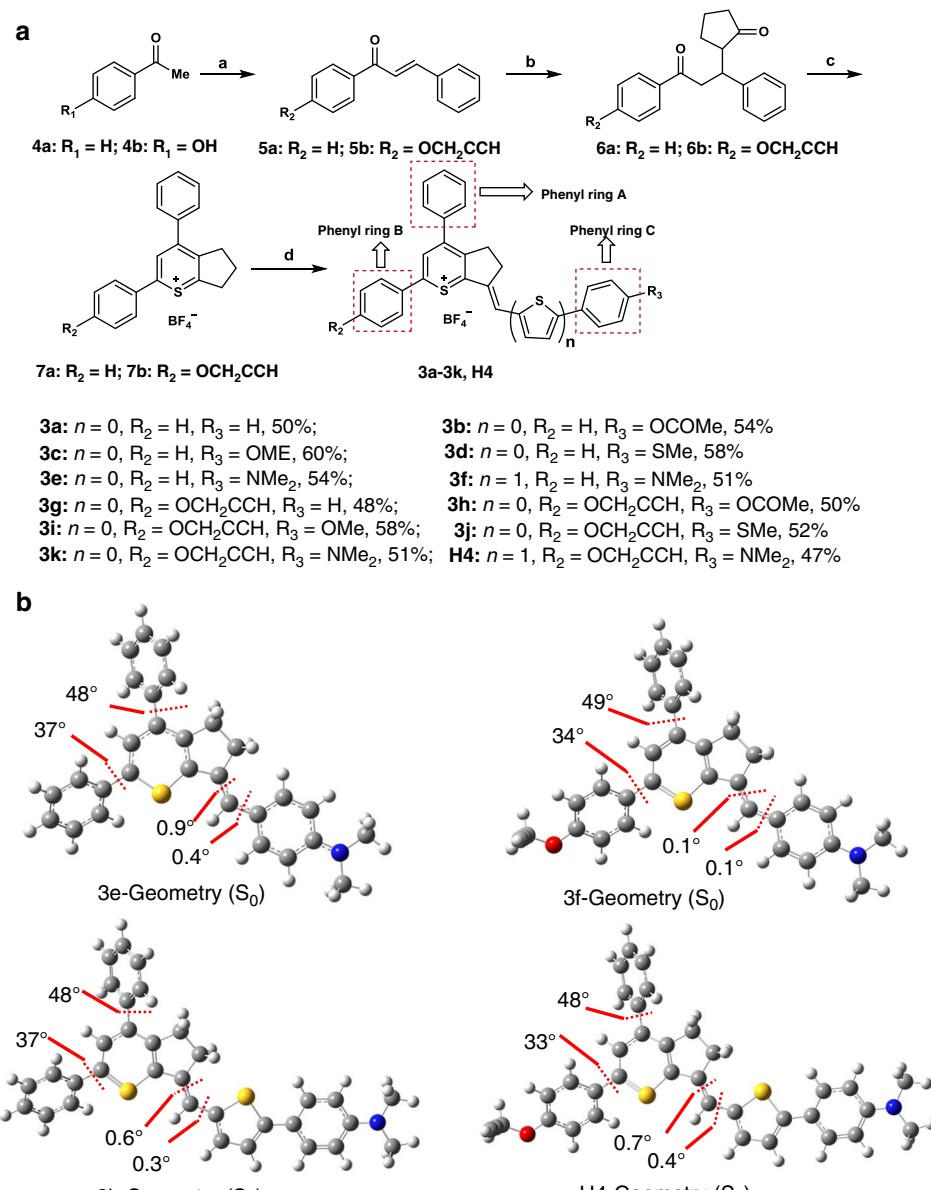

**Fig. 2 Synthetic route of 3a–3k, H4 and the calculated optimized geometries of 3e–3k, H4. a** Reagents and conditions: **a** (i) acetophenone, benzaldehyde, 3 M aqueous KOH, EtOH, 25 °C, 14 h, 5a: 80%; 1-(4-hydroxyphenyl)ethan-1-one: 82%; (ii) $K_2CO_3$, 3-bromoprop-1-yne, acetone, reflux 4 h, 5b: 97%; **b** (i) cyclopentanone, pyrrolidine, benzene, 100 °C, 4 h; (ii) compound 5, dioxane, reflux, 6 h, 6a: 68%; 6b: 66%; **c** thioacetic acid, boron trifluoride ether, ether, 60 °C, 2 h, 7a: 62%; 7b: 50%; **d** aldehydes, acetic anhydride, 70 °C, under microwave irradiation, 75 °C, 2 h, 47–60%. **b** Calculated optimized ground state ($S_0$) geometries of the molecules at the B3LYP/6-31 G (**d**) level (Gaussian 09, Revision D.09).

tetrafluoroborate 7a was obtained according to the literature[46]. Finally, subsequent condensation with various aldehydes under microwave irradiation resulted in the formation of the target molecule 3a–3f in moderately high yields of 50–60%. In order to introduce a modifiable group at the $R_2$ position instead of H, 3g–3k, and H4 with a methoxyethyne group were then synthesized according to the general strategy outlined in Fig. 2a.

DFT calculations with the B3LYP exchange functional employing 6–31G(d) basis sets were performed to identify the highest occupied molecular orbital (HOMO) and the lowest unoccupied molecular orbital (LUMO) of 3a–3k and H4 (Supplementary Table 1, Fig. 2)[48]. From the calculation results, we found that the $E_{gap}$ of compounds 3a–3f without a methoxyethyne unit were similar to that of 3g–3k, H4 with a methoxyethyne unit, which indicated that the presence of the methoxyethyne group did not have much effect on $E_{gap}$. The electron density of the HOMO of

3a–3f was mainly localized on the phenyl ring C with a minor contribution from positively charged sulfur ions, while the LUMO was primarily centered on the thiopyrylium unit, contributed by the p orbital on the sulfur ion and the phenyl ring A (Fig. 2b). The lower $E_{gap}$ of 3e and 3k was 2.07 and 2.06 eV, which can be attributed to the increased internal charge transferred by the introduction of a stronger electron-donating moiety of N,N-dimethylaniline group. However, the $E_{gap}$ of 3e and 3k was much higher than that of CH1055 (1.5 eV) with a typical NIR-II optical $E_{gap}$ based on previous results. Molecular engineering principles tell us that, to realize a lower $E_{gap}$, the acceptor requires a higher LUMO energy level. Meanwhile, a higher HOMO energy level is also needed to reduce the $E_{gap}$ value. The D–A-type fluorophore 3 can be further modified to achieve maximum emission within the NIR-II region by extending the conjugation with the electron-rich thiophene

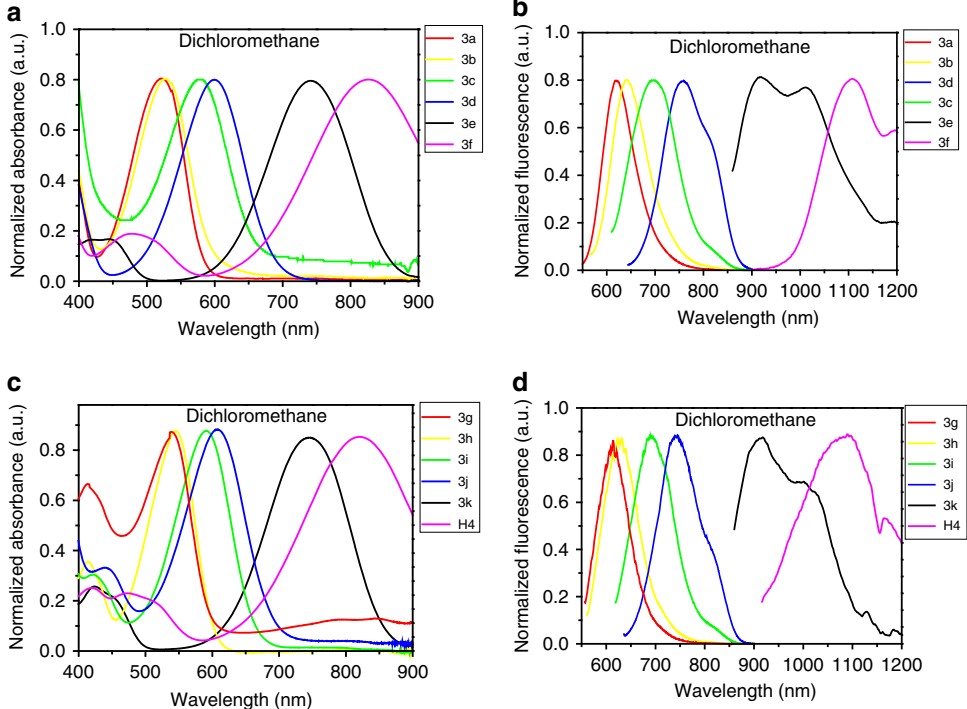

**Fig. 3 Spectroscopic properties of 3a–3k and H4. a** Absorption wavelength and **b** emission wavelength of 3a–3f in dichloromethane. **c** Absorption wavelength and **d** emission wavelength of 3g–3k and H4 in dichloromethane.

spacer in compounds 3f and H4, and using $N,N$-dimethylaniline as an electron donor to strengthen the intramolecular charge-transfer (ICT) effect. Moreover, 3f and H4 presented higher-lying calculated HOMO (−5.33 eV, −5.30 eV) and LUMO (−3.63 eV, −3.60 eV) levels, and showed lower band gaps (1.7 eV, 1.7 eV), as the electron-rich thiophene spacer significantly lowers the oxidation potential, resulting in a further bathochromic shift.

**Spectroscopic properties of 3a–3k and H4.** We then tested the spectroscopic properties of 3a–3f in dichloromethane as shown in Fig. 3a, b, and it was found that their emission wavelengths were almost the same as 3g–3k, H4 (Fig. 3c, d). These results fully demonstrated that the methoxyethyne group had minimal impact on the spectra of thiopyrylium dyes, and the methoxyethyne group did not act as a donor. The UV–Vis–NIR spectra of 3g–3k in THF, $CH_3CN$, Acetone, MeOH, EtOH, and DMSO were similar to the spectra in dichloromethane solution, indicating that the influence of these solvents on absorbance properties was almost negligible (Supplementary Fig. 1A)[49,50]. As shown in Supplementary Fig. 1, the magnitudes of the Stokes shift were consistent with the relative electron-donating abilities of the substituents on $R_3$. Increased electron density of donors was consistently accompanied by a shift in $\lambda_{ex}$ and $\lambda_{em}$ to longer wavelengths. If we considered compound 3a or 3g with unsubstituted benzene at $R_3$ as a reference, introduction of electron-donating substituents at the $R_3$ position such as OAc (3b, 3h), OMe (3c, 3i), and SMe (3d, 3j) led to a steady red-shift of fluorescence toward NIR-I wavelengths by 15–130 nm up to 31% fluorescence quantum yield. Interestingly, photoluminescence excitation mapping performed on 3e and 3k in dichloromethane showed maximum absorption peaks at ~741 and ~743 nm, and maximum emission peaks at ~917 and 918 nm, with a tail extending into the NIR-II region. Dimethylamino-substituent at the $R_3$ position induced an additional strong bathochromic shift of maximum absorption and fluorescence by enhanced electron-donating ability and resonance zwitterionic structure of $N,N$-

dimethylaniline substituted 3e and 3k. Quantum yields of 3a–3k were measured in dichloromethane (Supplementary Fig. 2 and Supplementary Table 2).

The UV–Vis-NIR absorption bands of 3f and H4 were at 580–1100 nm in $CH_2Cl_2$ due to the formation of a strong charge-transfer structure between D and A units, while the fluorescence emission spectra of 3f and H4 demonstrated peak emission wavelength at ~1100 nm (Fig. 3). Furthermore, NIR-II signals of H4 were systematically investigated under various solvents and acid conditions such as $CH_3CN$, DMSO, MeOH, EtOH/HCl, inorganic acids, organic acids, and Lewis acids (Fig. 4). The absorbance of the H4 solution remained unchanged under various organic solvents and acidic conditions. The absorbance and fluorescence increased linearly with the increased concentration of H4 (Fig. 4g, h), suggesting no aggregation by accumulation over a wide range of conditions. However, H4 exhibited a broad peak with significant quenching of emission in polar solvents, and increasing solvent polarity generally resulted in the shift of the emission spectra to longer wavelengths. H4 exhibited high photostability compared to ICG, with negligible decay under continuous excitation for 40 min (Fig. 4i) in DMSO. The quantum yield of H4 was 2.01% under 785 nm excitation in dichloromethane, measured against an IR-26 reference (0.5% quantum yield, Supplementary Fig. 3).

**In vitro evaluation.** Osteosarcoma is the third most common malignant tumor and accounts for more than 60% of all malignant bone tumors in children and adolescents[51]. Surgical excision of all primary and metastatic tumors remains the essential strategy behind osteosarcoma treatment. However, current diagnostic tools allow detection of only 8–15% of bone or lung metastatic patients[52,53]. Thus, it is urgent to develop new theranostic strategies for early diagnosis, therapy, and monitoring responses to osteosarcoma therapies[54,55]. The oligopeptide PPSHTPT (PT) was first designed to mimic the properties of the natural protein osteocalcin in vivo and possessed a high affinity

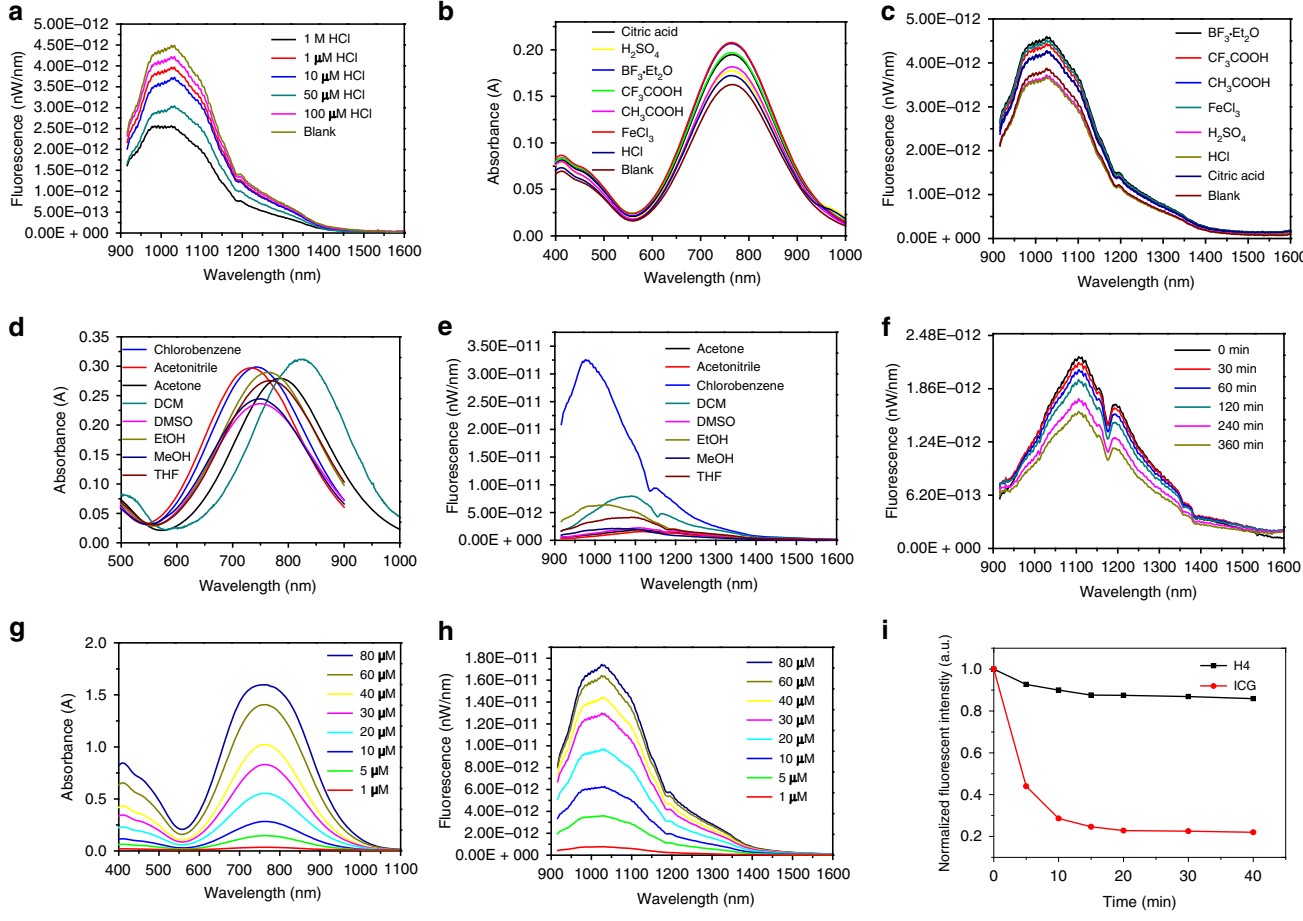

**Fig. 4 Spectroscopic properties of H4. a** Fluorescent emission of H4 (10 μM) in EtOH with different concentrations of HCl. **b** Absorbance and **c** fluorescent emission of H4 in different acids. **d** Absorbance and **e** fluorescent emission of H4 in 10 μM different solvents. **f** Fluorescent intensity of H4 in DMSO at different times. **g** Absorbance and **h** fluorescent emission of H4 in EtOH with different concentrations. **i** Compared stability of H4 with ICG in DMSO under continuous 808 nm laser irradiation.

and specificity for osteosarcoma cell lines (e.g., 143B cells)[56]. H4-PEG-PT was further synthesized through direct amidation and Cu-catalyzed azide-alkyne cycloaddition (Fig. 5a). Azide-PEG$_{16}$-COOH was first conjugated with an amine group of PT to give PEG-PT, then H4 was reacted with PEG-PT via Cu-catalyzed click reaction to obtain H4-PEG-PT. Normalized UV–vis spectra showed that H4-PEG-PT possessed a maximum absorption peak at ~800 nm and a maximum emission peak at ~1050 nm (Fig. 5g and Supplementary Fig. 3). Meanwhile, under ~850 nm excitation, H4-PEG-PT also showed an intense anti-Stokes FUCL signal with a maximum emission peak ~580 nm (Fig. 5h). H4-PEG-PT readily formed supramolecular assemblies in water with the average length of $180.0 \pm 13$ nm and the average width of $48 \pm 15$ nm as determined by transmission electron microscopy (TEM, Fig. 5i). The NIR-II quantum yield of H4-PEG-PT was $0.1 \pm 0.03\%$ in water. H4-PEG-PT also exhibited high photostability in PBS and DMEM medium for 60 min (Fig. 5e). Furthermore, fluorescent signals of hydroxyapatite (HA) precipitates were enhanced[57] when HA (10 mg) was incubated with different concentrations of H4-PEG-PT (4, 8, 16, 32, and 64 μM) for 4 h. H4-PEG-PT has demonstrated a high affinity for HA in vitro in bone-binding assays while much lower-intensity fluorescent signals were detected in HA precipitates with various concentrations of H4 without the targeting ligand PT (Fig. 5b, c). In addition, the binding ability of H4-PEG-PT to osteosarcoma was further confirmed by co-incubation of 143B cells with H4-PEG-PT. Much higher fluorescence was observed in the non-blocking

experiment (left tube, Fig. 5d) while negligible fluorescence was detected in the blocking experiment (right tube, Fig. 5d). The toxicity of H4-PEG-PT was determined in vitro with an MTT assay on 143B and L929 cells at different concentrations (2, 4, 8, 16, and 32 μM) (Fig. 5f). These results indicate that an aqueous soluble, photostable, and biocompatible NIR-II fluorescence probe H4-PEG-PT, is suitable for osteosarcoma imaging.

**FUCL Cellular localization of 3k-PEG, H4-PEG, and H4-PEG-PT**. Colocalization experiments involving Mito-Tracker Red (MTR) or Mito-Tracker Green (MTG) were conducted to confirm the intracellular distribution of the synthesized compounds. Specifically, the subcellular localization of 3k-PEG, H4-PEG, and H4-PEG-PT in 143B cells was determined by co-staining with MTG and then imaged under confocal microscope, while the subcellular distribution of 3j-PEG was observed by co-staining with MTR (Supplementary Movie 1 and Supplementary Movie 2). As shown in Fig. 6a–e, all of the fluorescence images for 3j-PEG, 3k-PEG, H4-PEG, or H4-PEG-PT with different concentrations were found to overlap well with that of commercial mitochondrial dyes (MTR or MTG), suggesting the mitochondria-targeted ability of those synthesized dyes. Moreover, it was found that the fluorescent intensity of Mitotracker Green at concentrations lower than 10 nM was too weak to be detected while 1 nM of H4-PEG-PT still presented clear and strong fluorescent signals under the same experimental conditions. NIR-II fluorescent confocal studies (Fig. 6 and Supplementary Fig. 6) demonstrated that

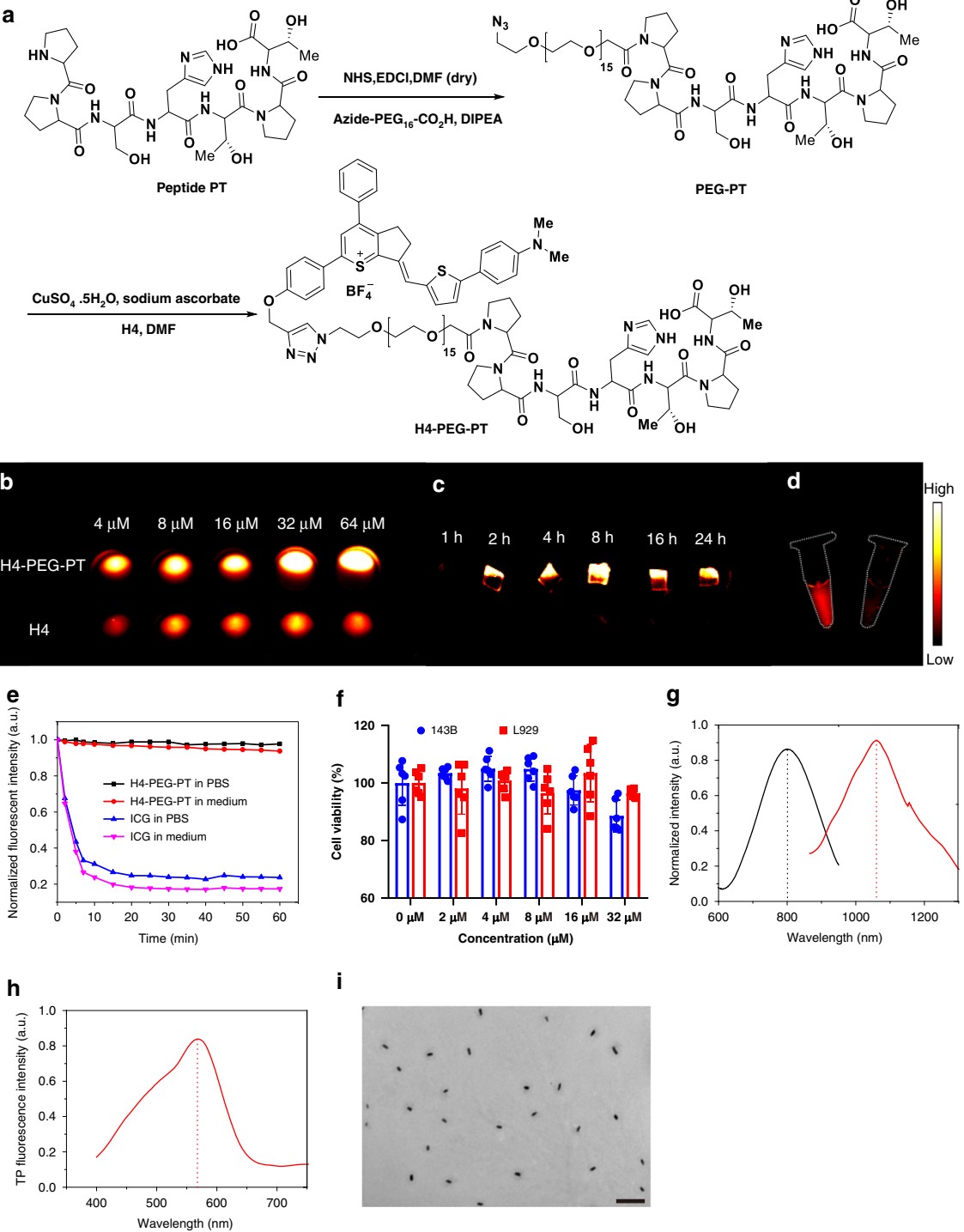

**Fig. 5 In vitro evaluation of H4-PEG-PT. a** Synthesis of H4-PEG-PT via amidation and click reaction. **b** Fluorescence image of hydroxyapatite (HA) binding of H4 and H4-PEG-PT with different concentrations (4, 8, 16, 32, and 64 μM). **c** Fluorescence image of cartilage slices incubated for various time points with H4-PEG-PT. **d** NIR-II signals of 143B cells labeling by H4-PEG-PT (left) and H4-PEG-PT and excess PT as the blocking agent (right) under an 808 nm excitation (1000 LP and 100 ms). **e** Compared stability of H4-PEG-PT vs. ICG in different media under continuous laser irradiation. **f** Cellular toxicity of H4-PEG-PT with different concentrations (2, 4, 6, 8, 16, and 32 μM) in 143B and L929 cells (data are presented as mean values ± SD, derived from $n = 6$ biologically independent samples). **g** Absorption and emission wavelength of H4-PEG-PT under 785 nm excitation in water. **h** Frequency upconversion luminescence (FUCL) of H4-PEG-PT under 850 nm excitation in water. **i** TEM image of H4-PEG-PT (scale bar: 500 nm). The results in (**i**) are representative of three independent experiments.

3j-PEG, 3k-PEG, H4-PEG, and H4-PEG-PT were taken up by mitochondria across the inner membrane and achieved mitochondria-targeted via the delocalized lipophilic cation thiopyrylium salt instead of the alkyltriphenylphosphonium moiety.

In summary, 3k-PEG, H4-PEG, and H4-PEG-PT were shown to be the first D–A type FUCL thiopyrylium mitochondria-targeted fluorophores with highly desirable characteristics including mitochondria-targeted and low working concentration.

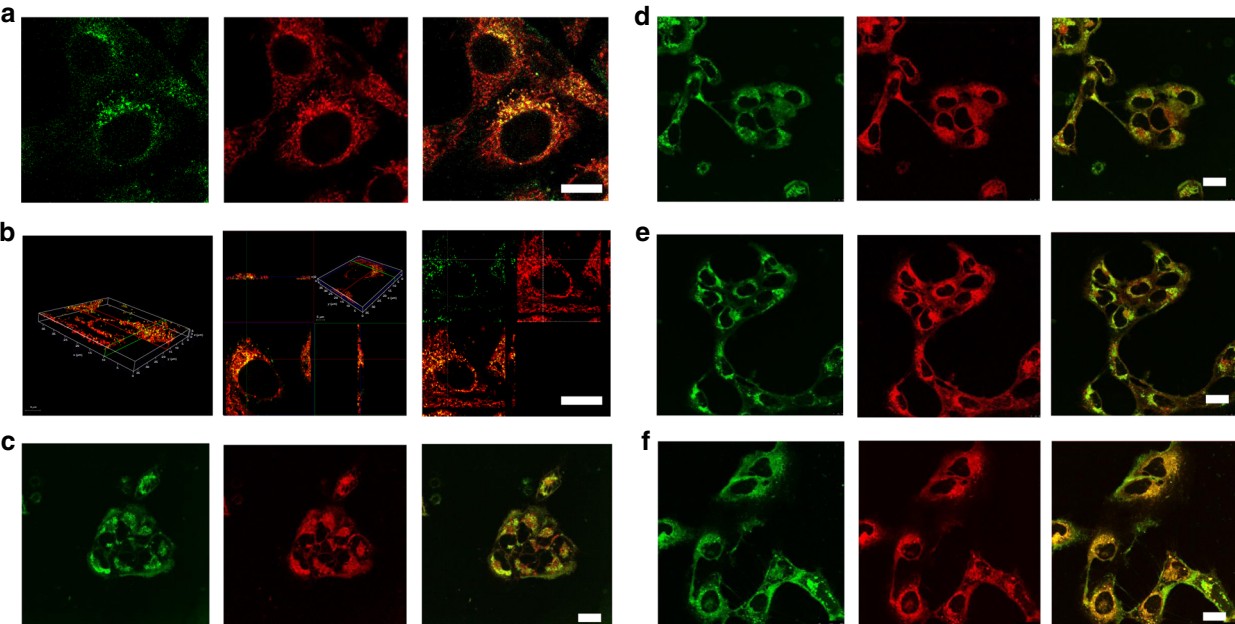

**Fig. 6 Mitochondrial localization. a** Fluorescent images of 143B cells incubated with 3j-PEG (10 μM) for 8 h (left); with Mito-tracker red (middle) and merge images of them (right). $\lambda_{em}$: 590–650 nm, $\lambda_{ex}$: 594 nm (Mito-tracker red); $\lambda_{em}$: 490–540 nm, $\lambda_{ex}$: 488 nm (3j-PEG). **b** Merge images of 3j-PEG and Mito-tracker red in 3D fluorescence images of mitochondrial localization. **c–e** Fluorescent images of 143B cells incubated with Mito-tracker green (left), with **c** 3k-PEG (1 μM) or **d** H4-PEG (1 μM), or **e** H4-PEG-PT (1 μM) for 6 h (middle) and merge images them (right), respectively. $\lambda_{em}$: 495–526 nm, $\lambda_{ex}$: 488 nm (Mito-tracker green); $\lambda_{em}$: 610–660 nm, $\lambda_{ex}$: 514 nm (3k-PEG, H4-PEG, and H4-PEG-PT). **f** Fluorescent images of 143B cells incubated with Mitochondria-GFP (left), H4-PEG-PT (middle), and merge images of them (right). $\lambda_{em}$: 495–526 nm, $\lambda_{ex}$: 488 nm (Mitochondria-GFP); $\lambda_{em}$: 610–660 nm, $\lambda_{ex}$: 514 nm (H4-PEG-PT). Scale bar: 10 μm. Results in (**a–f**) are representative of three independent experiments.

**In vivo imaging**. The 143B mice tumor model was established by orthotopic injection of green fluorescent protein (GFP)-transfected 143B cells. Orthotopic tumor growth was monitored by GFP luminescence and X-ray imaging until the proximal tibia of the joints were phagocytosed by 143B cells (Fig. 7a). H4-PEG-PT was then intravenously injected (200 μg, 200 μL) into 143B tumor-bearing mice. From NIR-II imaging, 143B tumors were clearly discernible from the surrounding background tissues from 6 to 48 h (Fig. 7b, 1000 LP, 250 ms). Compared with the normal group, lower fluorescent signals were observed at all time points in the blocking group. The NIR-II intensity ratios (T/N) for H4-PEG-PT were shown in Supplementary Fig. 7, and its T/N ratio reached 4.57 ± 0.15 at 12 h. The targeting specificity of H4-PEG-PT for osteosarcoma was confirmed by the blocking experiment. After co-injection of free PT peptide (2 mg) with H4-PEG-PT for NIR-II imaging, tumor fluorescent signals were significantly reduced at all time points. H4-PEG-PT distribution in major organs at 48 h was evaluated by an ex vivo biodistribution study (Supplementary Fig. 8). The predominance of liver accumulation suggested that the hepatobiliary system was the primary site for the clearance of H4-PEG-PT. The half-life of H4-PEG-PT in the blood circulation was assessed by quantitative determination of fluorescent intensity in the blood at different time points, as shown in Supplementary Fig. 9. The time-dependent relative fluorescent intensity in blood was fitted with a first-order exponential decay to evaluate the blood half-life. The equation of the pharmacokinetic curve was $y = 0.061*\exp(-X/0.397) + 0.00994$ ($R^2 = 0.989$) and the blood half-life ($t_{1/2}$) was found to be ~0.4 h. In addition, tumor uptake of H4-PEG-PT reached a maximum at 12 h. Histopathological studies were also carried out by ex vivo NIR-II fluorescent imaging and hematoxylin-and-eosin (H&E) tissue staining (Supplementary Fig. 9).

**In vitro photothermal efficacy**. Photothermal therapy in the NIR optical window has received growing attention on account of its convenient and efficient use of NIR light as well as its considerable advantages of deeper tissue penetration and higher maximum permissible exposure to laser[58–61]. H4-PEG-PT was exposed to an 808 nm laser to investigate photothermal performance via a mitochondrial pathway at various concentrations. Rapid photothermal heating occurred upon irradiation at 1.6 W cm$^{-2}$ even at the dose of 32 μM (Fig. 8a). When the concentration of H4-PEG-PT was increased to 64 μM, the temperature increased significantly under laser irradiation by varying the power density (1.4, 1.6, 1.8, and 2.0 W cm$^{-2}$) as shown in Supplementary Fig. 10A. Exposure to 808 nm laser at 2.0 W cm$^{-2}$ for 12 min, the temperature of H4-PEG-PT was increased to 57 °C while the temperature of the pure water was increased only slightly under the same laser irradiation. To further investigate the photostability of H4-PEG-PT, it was subjected to repeated heating-cooling cycles for 80 min (Fig. 8b), indicating excellent stability under 808 nm laser irradiation (2.0 W cm$^{-2}$). Then, 143B cells were irradiated with an 808 nm laser for 4 min (1.8 and 2 W cm$^{-2}$) after incubation with different concentrations of H4-PEG-PT. Cells were obviously destroyed when the concentration of H4-PEG-PT reached concentrations of 32 μM or higher (Fig. 8c). However, the control groups of H4-PEG-PT without 808 nm irradiation showed almost 100% cell viability. The photothermal conversion efficiency of H4-PEG-PT was 18% ($\eta \approx 18\%$) (Supplementary Fig. 10), indicating that H4-PEG-PT could convert 808 nm light energy into heat energy efficiently and rapidly.

To explore the underlying mechanisms of H4-PEG-PT under NIR laser irradiation against 143B cells, its effects on mitochondrial function and apoptosis were investigated (Fig. 8d–f). Mitochondrial functioning was first assessed by determining the

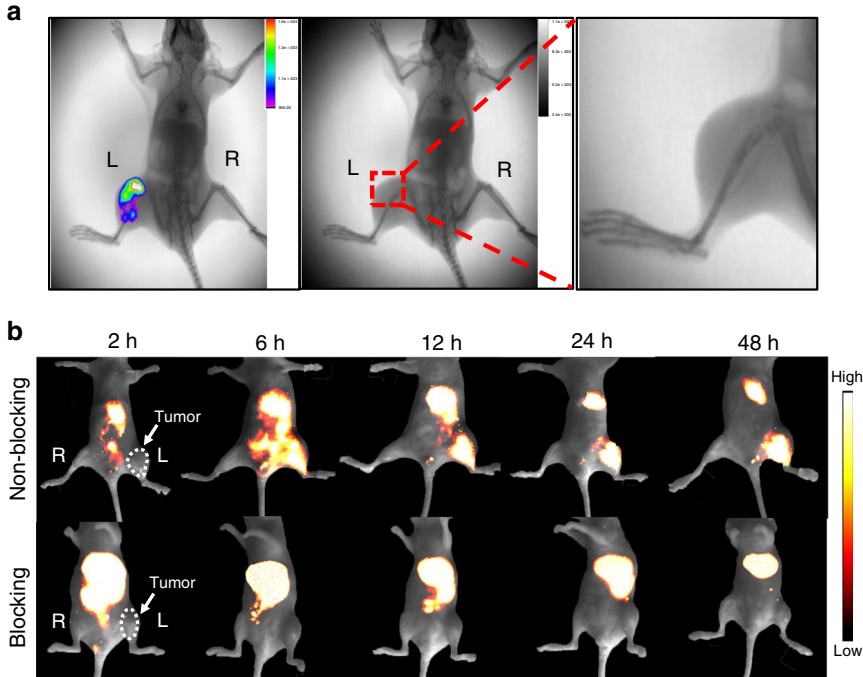

**Fig. 7 In vivo imaging of osteosarcoma. a** Merge of X-ray and bioluminescence imaging of orthotopic 143B tumor in the nude mouse (left), X-ray imaging of orthotopic 143B tumor in the nude mouse (middle) and enlarged images of proximal tibia of the joint (right) (prone position) ($n = 3$ biologically independent mice). **b** NIR-II signals of H4-PEG-PT in 143B tumor mice (top) and the blocking group (bottom) under 808 nm excitation (1000 LP, 3.5 W, 250 ms) (supine position) ($n = 3$ biologically independent mice).

levels of ATP in 143B cells (Fig. 8d). Pretreatment with H4-PEG-PT or 808 nm irradiation, the levels of ATP were similar to those in control cultures. H4-PEG-PT under NIR irradiation treatment significantly decreased 143B cellular ATP generation to ~5.6% (Fig. 8d). JC-1 assay was further used to analyze mitochondrial membrane potential ($\Delta\Psi_m$). The ratio of J-aggregates to monomers of H4-PEG-PT (60 μM) or NIR irradiation treatments for up to 3 min had no significant decrease. Treatment with H4-PEG-PT (60 μM) under NIR irradiation or carbonyl cyanide *m*-chlorophenylhydrazone (CCCP) reduced the ratio of J-aggregates to monomers of 143B tumor cells to 25% and 63%, respectively (Fig. 8e). The change in the permeability of mitochondrial membranes is a significant event in the apoptosis process, which is controlled by mitochondrial permeability transition (MPT). In our experiment, calcein-AM cobalt assay was used to identify MPT. The experimental results showed that, after 143B cells were treated with H4-PEG-PT under an 808 nm irradiation for 3 min, the calcein fluorescence intensity of 143B cells was much lower than that of $H_2O_2$-induced group (Fig. 8f). These results revealed that H4-PEG-PT under NIR irradiation would cause serious mitochondrial physiology dysfunction against cancerous 143B cells, while H4-PEG-PT or NIR irradiation treatment alone could not damage cells.

During apoptosis, Cytochrome c (Cyto c) was released from mitochondria to the cytosol and further activated the caspase-dependent apoptosis pathway, which committed the cell to the death process[62]. In order to further understand the mechanisms of cell toxicity and mitochondrial-targeted properties of 3k-PEG, H4-PEG, and H4-PEG-PT, tests of subcellular distribution of Cyto c from the immunofluorescence technique and expression levels of caspase-3 and caspase-9 as well as their activated fragments were carried out by western blotting analysis. 3k-PEG, H4-PEG, and H4-PEG-PT were well retained within mitochondria even after cell fixation and permeabilization at 1 nM, and red fluorescence from Cyto c was clearly observed in all cells in

combination with 808 nm laser irradiation. Notably, highest fluorescence of Cyto c was observed in the group treated with H4-PEG-PT and 808 nm laser irradiation (Fig. 8q, Supplementary Fig. 11). These results further demonstrated the mitochondrial-targeting photothermal effect of the synthesized NIR-II probes. It is well-known that caspase-9 and caspase-3 are important members of the cysteine-aspartic acid protease family, which regulates cellular apoptosis, and their sequential activation plays a key role in the mitochondria-mediated cell apoptosis pathway[63]. Western blot analysis showed that H4-PEG-PT dramatically reduced caspase-3 and caspase-9 protein levels of 143B cells under 808 nm irradiation (1 W cm$^{-2}$, Fig. 8k). Afterwards, the expression of catalytically active caspase-3 and cleaved caspase-9 were significantly increased. Besides, the increase of caspase-3/8/9 activities was also observed in the H4-PEG-PT group under NIR irradiation by the caspase multiplex activity assay kit (Fig. 8g–i). Our findings further supported that cellular apoptosis of H4-PEG-PT was triggered by the intrinsic mitochondrial pathway. Upon death stimuli, different apoptogenic factors such as apoptosis-inducing factor (AIF), Endonuclease G (EndoG) and second mitochondria-derived activator of caspase/direct IAP binding protein with low pI (Smac/Diablo) are released during the early stages of apoptosis. These three characteristic proteins of mitochondrial apoptosis were investigated by western blot. From Fig. 8j, the AIF and EndoG expression levels in the H4-PEG-PT group with NIR irradiation were significantly increased compared with that of the same group without NIR light irradiation, control group or NIR laser group. Western blotting results indicated that there was burst release of AIF, EndoG and Smac/DIABLO from mitochondria into the cytoplasm with H4-PEG-PT under irradiation, suggesting the disruption of mitochondria and cellular metabolism. NIR light-triggered ROS generation of H4-PEG-PT under 808 nm laser irradiation was then evaluated by 1,3-diphenylisobenzofuran (DPBF), dichloro-dihydro-fluorescein diacetate (DCFH-DA) and MitoSOX assay (Fig. 8l–p, r and

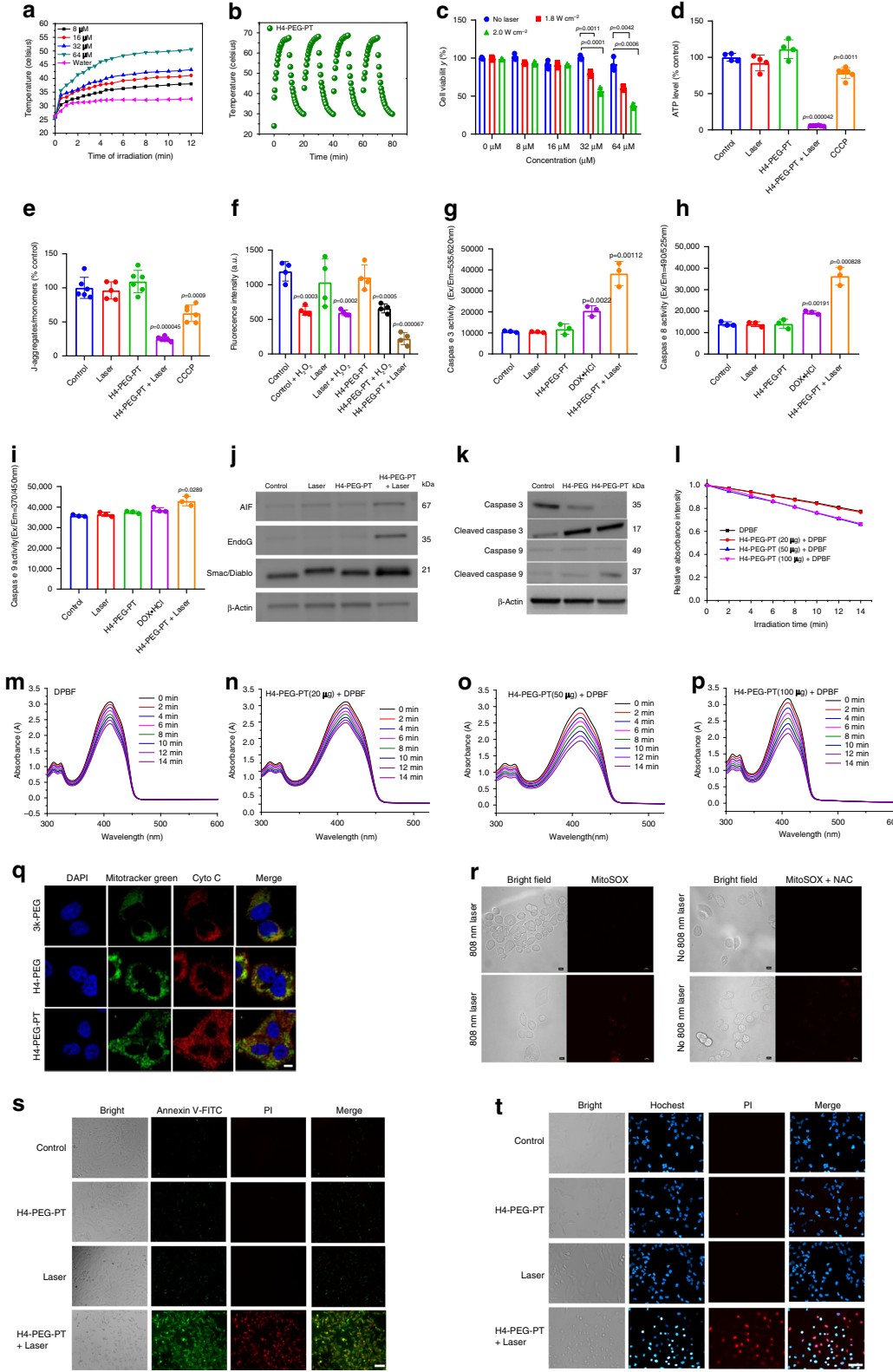

Supplementary Fig. 12)[64,65]. No significant amount of ROS in vitro was observed in any group of H4-PEG-PT under 808 nm irradiation. This result is a positive finding regarding photothermal application of H4-PEG-PT since the concomitant production of ROS during PTT has been proven to trigger detrimental side-effects to cells[66,67]. FITC-annexin V and propidium iodide (PI) assay is usually used to identify early and late-stage apoptotic versus necrosis. After an 808 nm laser irradiation for 3 min, not only FITC signals were obviously emerged from the cell membrane but also PI signals were visible in the group of H4-PEG-PT with laser irradiation. Then Hochest/ PT staining assay was undertaken to examine the engagement of apoptosis VS necrosis or mixed paths. 143B cells treated with H4-PEG-PT under irradiation clearly demonstrated staining

**Fig. 8 In vitro photothermal efficacy and photoinduced cytotoxicity. a** Photothermal heating curves of H4-PEG-PT with different concentrations (200 μL) at a laser power of 1.6 W cm$^{-2}$. **b** H4-PEG-PT (75 μM, 200 μL) over several ON/OFF cycles involving irradiation with an 808 nm laser (2.0 W cm$^{-2}$) for 80 min followed by passive cooling. **c** Cell viability of 143B cells after incubation with various concentrations of H4-PEG-PT and exposure to different laser power densities ($n = 3$ biologically independent samples). **d** Cellular ATP production of 143B cells treated with H4-PEG-PT w/o 808 nm laser irradiation, CCCP was the positive control ($n = 4$ or 5 biologically independent samples). The ATP level of control was 100%. **e** Mitochondrial membrane potential quantified by JC-1 fluorescence assay of 143B cells treated with H4-PEG-PT w/o 808 nm laser irradiation, CCCP was the positive control ($n = 5$ or 6 biologically independent samples). **f** Mitochondrial permeability transition (MPT) study of H4-PEG-PT after 808 nm laser irradiation using calcein-cobalt assay. $H_2O_2$ treatment is the positive control ($n = 4$ biologically independent samples). **g–i** Caspase-3/8/9 activities measured by caspase multiplex activity assay kit in different groups ($n = 3$ biologically independent samples). **j** Representative western blot analysis of AIF, EndoG, and Smac/Diablo in different groups. **k** Representative western blot analysis of Caspase-9, Caspase-3, cleaved Caspase-3 (or active Caspase-3), and cleaved Caspase-9 (or active Caspase-9) after photothermal therapy under 808 nm excitation for H4-PEG and H4-PEG-PT, respectively. **l** Relative absorption intensity at 410 nm of DPBF as a functional of 808 nm light radiation time (DPBF alone, incubated with H4-PEG-PT with different concentrations, respectively). **m–p** Absorption spectra of DPBF solution (DPBF alone, or incubated with H4-PEG-PT with different concentrations) under 808 nm irradiation for different time. **q** Confocal images of 143B cells at 12 h after 3k-PEG, H4-PEG, and H4-PEG-PT treatments in combination with 808 nm laser irradiation. Mitochondria were stained with MitoTracker Green, whereas Cyto c (red fluorescence) was detected using immunofluorescence technique. Scale bar: 10 μm. **r** Confocal fluorescence microscopic images of 143B cells incubated with H4-PEG-PT and MitoSOX according to the treatment variables. Scale bar: 10 μm. **s, t** Apoptosis or necrosis of 143B cells determined by (**s**) Annexin V-FITC/PI staining or (**t**) Hochest/PI staining (scale bar: 50 μm). Statistical significance was calculated with two-tailed Student's *t* test (**c–i**). Data are presented as mean values ± SD (**c–i**). Results in (**j**), (**k**), (**q–t**) are representative of three independent experiments.

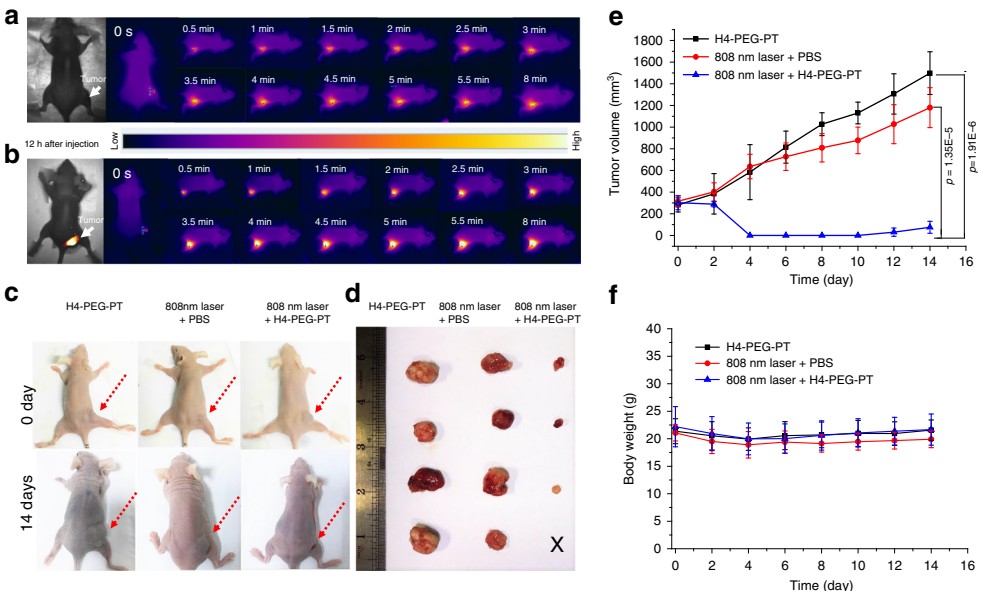

**Fig. 9 In vivo photothermal evaluation of H4-PEG-PT against osteosarcoma. a** PBS and **b** H4-PEG-PT solution injected to tumor bearing mice under 808 nm laser irradiation (1.5 W cm$^{-2}$). **c** Representative photographs of mice after different treatments: H4-PEG-PT only, PBS under 808 nm laser irradiation, and H4-PEG-PT exposed to 808 nm irradiation, respectively. Red dashed arrows indicated tumor. **d** Photographs of tumor tissue obtained in different groups after 14 d treatment, respectively. **e** Tumor volumes and **f** body weight of different groups of 143B tumor-bearing mice after treatment ($n = 4$ biologically independent mice). Data are presented as mean values ± SD (**e, f**). Statistical significance was calculated with two-tailed Student's *t* test (**e**).

indicative of apoptosis (bright-blue particles). Because annexin V could stain early-stage apoptotic cells while PI stains dead cells or late-stage apoptotic cells, and Hochest could stain the nuclei of apoptotic cells with a bright-blue fluorescence, these two results showed that the cells treated with H4-PEG-PT under irradiation were engaged with both apoptosis and necrosis (Fig. 8s, t).

**In vivo photothermal therapy.** Encouraged by the findings of excellent photothermal properties in vitro, the effects of in vivo photothermal therapy in 143B tumor-bearing mice under an 808 nm NIR laser were also studied. H4-PEG-PT (200 μL, 400 μg) or PBS (200 μL) were intravenously injected into 143B tumor-bearing mice respectively ($n = 4$ per group). The right tumor regions of the mice were irradiated with an 808 nm laser (1.5 W

cm$^{-2}$) continuously for 8 min after 12 h i.v. post-injection. The tumor surface temperature rapidly increased from 33.0 to 67.8 °C in 8 min (Fig. 9b), while the normal skin exhibited only a slight temperature increase (Fig. 9a). Tumor sizes and body weights were measured every 2 d as shown in Fig. 9c–f. No significant body weight loss was observed in any group during the photothermal therapy process. After 14 d of treatment, mice were sacrificed and major organs collected for histopathological studies with H&E staining. No obvious injury or inflammatory lesion was observed in any major organ, as shown in Supplementary Fig. 13. Overall, both in vitro and in vivo photothermal studies showed that H4-PEG-PT efficiently converts 808 nm laser light into heat with negligible production of ROS, and exhibits effective photothermal efficacy against 143B tumors without a significant adverse effect.

## Discussion

The fluorophores in this work composed of a series of novel red-shifted thiopyrylium cores from heterocycles 7a and 7b. After theoretical calculation and optical structure–property relationship study, D–A thiopyrylium dye H4 derived by the conjugation with the electron-rich thiophene spacer and N,N-dimethylaniline substituent was discovered, and exhibited dramatic bathochromic shifts of π–π* transitions in the NIR-II wavelength region with emission wavelengths at ~1100 nm. In addition, a water-soluble and biocompatible upconversion NIR-II probe H4-PEG-PT was successfully constructed via amidation and click chemistry based on the thiopyrylium dye H4 and an osteosarcoma-targeting oligopeptide PPSHTPT (PT), demonstrated specifically mitochondria-targeted imaging of osteosarcoma in vitro and in vivo. Moreover, the synthesized fluorophore functioned as a photothermal agent by actively targeting tumor tissues and mitochondria to selectively kill cancer cells. It has been demonstrated that D–A thiopyrylium dyes such as H4-PEG-PT could serve as effective mitochondrial-targeting photothermal agents without ROS effects and achieve markedly enhanced antitumor efficacy. In addition, H4-PEG-PT showed no observable toxicity without laser irradiation, indicating their safety. Besides, H4-PEG-PT showed super mitochondria-targeted FUCL bioimaging in live cancer cells. Notably, the majority of commercially available mitochondrial dyes are suited for live samples. We developed a small-molecule imaging agent that can quickly and effectively image mitochondria in live or fixed cell samples with subcellular resolution at a very low working concentration (1 nM). Western blot analysis showed that both H4-PEG and H4-PEG-PT dramatically reduced caspase-3 and caspase-9 protein levels, which further suggested that their photothermal effect was triggered by a mitochondria-induced apoptosis pathway. Thus, combining favorable clearance with biocompatibility, excellent photothermal therapy, as well as NIR-II tumor-imaging ability, H4-PEG, and H4-PEG-PT hold great promise for clinical translation. Importantly, chemical versatility makes H4-PEG and H4-PEG-PT suitable for various modes of tumor detection/imaging/image-guided surgery with high safety profile, subcellular resolution, and deep tissue penetration, and therapy by conjugating with different targeting ligands.

Taken together, a small-molecule organic fluorophore with cancer cell mitochondrial targeting, NIR-II upconversion imaging, and photothermal effects was developed. The design and synthesis of H4-PEG-PT will enrich and broaden the field of NIR-II and FUCL-based probes. It is hoped that this integrated small-molecule-based theranostic platform may become a practicable strategy for simultaneous cancer diagnostics, organelle targeting, therapeutics, and fluorescence-guided surgery, as well as postoperative monitoring.

## Methods

Information includes descriptions of thiopyrylium tetrafluoroborate 7, 3a–3k. H4, 3j-PEG-H4-PEG, H4-PEG-PT design and synthesis, bioconjugation, and purification are described in detail in SI Appendix.

**Quantum yield measurements of H4 and H4-PEG-PT**. In order to measure the relative quantum yield of H4 in chlorobenzene, IR-26 (0.5%) in 1,2-dichloroethane (DCE) was chosen as the reference. The quantum yield was calculated in the following formula:

$$QY_{sample} = QY_{IR26} \times \frac{Slope_{sample}}{Slope_{IR26}} \times \frac{n^2_{sample}}{n^2_{IR26}} \qquad (1)$$

Five different concentrations were chosen at or below OD 0.1 and the integrated fluorescence was plotted against absorbance for both IR-26 and H4. Comparison of the slopes led to the determination of the quantum yield of H4.

**Cell line and animal model**. L929 and HepG2 cells were purchased from the China Center for Type Culture Collection (CCTCC). 143B cells were obtained from Zhongnan Hospital of Wuhan University (Hubei, China), purchased from American Type Culture Collection (ATCC). 143B, L929, and HepG2 cells were cultured with Dulbecco's modified Eagle's medium (DMEM) (Gibco, Grand Island, NY, USA) supplemented with 10% fetal bovine serum (FBS) (Gibco, Grand Island, NY, USA) and 1% (v/v) penicillin-streptomycin at 37 °C in a humidified incubator containing 5% $CO_2$. The orthotopic 143B tumor models were established by intra-tibial injection of 143B cells (~$5 \times 10^6$ in 80 μL of PBS) into the left leg of 6-week-old female Balb/c nude mice (Beijing Vital River Laboratory Animal Technology Co., Ltd.) under anesthesia using isoflurane. The tumor-bearing mice were imaged when the tumor volume reached 400–800 $mm^3$ (about 4–6 weeks after inoculation).

**Cellular toxicity of H4-PEG-PT by MTT assay**. In vitro cytotoxicity studies of H4-PEG-PT on 143B cells and L929 cells were performed by using an MTT cytotoxicity assay. Cells were seeded in 96-well plates at densities between 5000 and 10,000 cells per well. The cells were incubated with 100 μL of fresh cell media containing H4-PEG-PT for 48 h. The final concentrations of H4-PEG-PT in the culture medium were fixed at 0, 2, 4, 8, 16, and 32 μM in the experiment. Then MTT (0.5 mg/mL) was added into each well (10 μL) in order to convert into formazan (purple). And the microplate was incubated at 37 °C for 4 h. After that, using DMSO to resolubilize the formazan. The absorbance was measured at 490 nm by Perkin Elmer VICTOR X4. The following formula was used to calculate the viability of cell growth: Cell viability (%) = (mean of absorbance value of treatment group/mean of absorbance value of control) × 100.

**Extracellular and intracellular ROS detection**. The generation of extracellular ROS was measured with a DPBF probe. Briefly, different concentrations of H4-PEG-PT (20, 50, 100 μg, containing 100 μl EtOH solution) were added into 3 mL of ethanol solution containing DPBF (10 mM), and then the solution was kept in the dark with magnetic stirring and irradiated by an 808 nm NIR laser for various time periods (0, 2, 4, 6, 8, 10, 12, and 14 min). The spectrum was collected at 410 nm absorption.

**Confocal laser scanning microscope (CLSM)**. 143B (50,000 cells) were seeded in round discs and incubated in 1 mL of DMEM medium (pH = 7.4) containing 10% FBS for 24 h. After replacing the medium with 1 mL of fresh medium, 3k-PEG, H4-PEG, and H4-PEG-PT were added, respectively, and the cells were allowed to incubate for 6 h. After removing the medium and subsequently washing with PBS buffer (pH = 7.4) thrice, mitotracker green was added to stain the mitochondria for 30 min. The cells were viewed under a FV1200 CLSM (Olympus) and Leica SPE.

**Detection of mitochondrial ROS formation**. 143B cells were seeded on 15 mm confocal dishes and allowed to stabilize for 24 h. The cells were then incubated with medium containing H4-PEG-PT (64 μM) for 6 h and washed three times with PBS. After 30 min incubation with or without N-acetylcysteine (5 mM) in 1 mL culture medium, the cells were irradiated with an 808 nm laser (1.0 W cm$^{-2}$) for 5 min. The amount of ROS generated was determined using the MitoSOX (1 μM) staining kit according to the manufacturer's instructions (Invitrogen). The results in 143B cells were monitored using a confocal fluorescence microscope. Fluorescence was collected using an excitation wavelength of 488 nm and recording the emission at 550–650 nm (red).

**Subcellular distribution of Cytochrome c (Cyto c)**. 143B cells were seeded onto an 8-well Chambered Coverglass system at a density of $2 \times 10^4$ cells per well and cultured for 24 h. 3k-PEG, H4-PEG, and H4-PEG-PT (32 μM) were added. To assess the subcellular distribution of Cyto c, the 143B cells treated with and without 808 nm laser irradiation were further incubated for additional 48 h and then 200 nM MitoTracker Green was added to stain the mitochondria. Afterward, the cells were fixed with 4% paraformaldehyde and then blocked with 1% BSA, 22.52 mg/mL glycine in PBST (PBS + 0.1% Tween 20) for 1 h. Subsequently, the cells were examined via the immunofluorescence technique using primary anti-Cyto c monoclonal antibody (Abcam, 1:1000) and secondary Alexa-647-conjugated goat anti-rabbit antibody (Life Technologies, 1:300). After staining with hoechst 33342 (Thermofisher Scientific, 1:10,000), the subcellular distributions of Cyto c in 143B cells under different treatments were observed by confocal microscopy (FLUO-VIEW FV1000, Olympus).

**Western blot analysis and antibodies**. Protein extracts were prepared using modified radioimmunoprecipitation assay lysis buffer (50 mM Tris-HCl pH 7.4, 150 mM NaCl, 1% NP-40 substitute, 0.25% sodium deoxycholate, 1 mM sodium fluoride, 1 mM Na$_3$VO$_4$, 1 mM EDTA), supplemented with protease inhibitor cocktail (Cell Signaling) and 1 mM phenylmethanesulfonyl fluoride or complete Mini protease inhibitor tablets (Roche). Equal amounts of protein, as determined with the bicinchoninic acid (BCA) protein assay kit (Pierce/Thermo Scientific) according to the manufacturer's instructions, were resolved on SDS-PAGE gels and transferred to nitrocellulose or polyvinylidene difluoride membrane. The blots were blocked with 5% nonfat dry milk or 3% BSA in TBST (50 mM Tris-HCl, pH 7.4 and 150 mM NaCl, and 0.1% Tween 20) and then incubated with appropriate

primary antibodies. Signals were detected with horseradish peroxidase (HRP)-conjugated secondary antibodies and an enhanced chemiluminescence (ECL) detection system (Pierce). For biochemical analyses, the following antibodies were used: Recombinant Anti-Smac/Diablo antibody (Abcam, 1:1000, ab32023); anti-EndoG antibody (Abcam, 1:1000, ab9647); Recombinant Anti-AIF antibody - Mitochondrial Marker (Abcam, 1:1000, ab32516); Recombinant Anti-Cytochrome C antibody, (Abcam, 1:1000, ab133504), Secondary Alexa-647-conjugated Goat anti-Rabbit Antibody (Life Technologies, 1:300, A27040); HRP conjugated Goat Anti-Rabbit IgG H&L (HRP) (Abcam, 1:2000, ab6721); anti-caspase-3 (Cell Signaling Technology, 1:1000, #9662S), anti-cleaved caspase-3 (Cell Signaling Technology, 1:1000, #9664S), anti-cleaved caspase-9 (Cell Signaling Technology, 1:1000, #9509S), anti-caspase-9 (Cell Signaling Technology, 1:1000, #9504S), and anti-beta actin antibody (Abcam, 1:2000, ab179467).

**Detection of the tumor by bioluminescence imaging**. Tumor-bearing mice that implanted 143B cells transfected with green fluorescent protein (GFP) were subjected to anesthesia with 10% chloral hydrate, followed by whole-body real-time bioluminescence imaging with In Vivo Xtreme imaging system (XTEREME BI, Bruker). BLI signal was collected using the following settings: 480 nm excitation and 535 nm emission; exposure time: 1 s; Bin: $1 \times 1$ pixels; FOV: 7.2 cm; fStop: 1.1; focal plane: 0 mm.

**In vitro bone-targeted assay**. In order to evaluate the binding ability of H4-PEG-PT and H4, various amount of the probes (2, 4, 8, 16, 32, and 64 μM, 500 μL) were incubated with HA (10 mg), respectively, for 4 h at room temperature under mild stirring. Then, the mixture was centrifugated at 1200 rpm for 10 min and washed with 8-fold excess of PBS three times. The NIR-II images were collected by a Series II 900/1700 In Vivo Imaging System (Suzhou NIR-Optics Co., Ltd., China).

**In vivo NIR-II fluorescence imaging of tumors**. Two-hundred microliter portions of H4-PEG-PT (200 μg) was injected intravenously into tumor-bearing nude mice. NIR-II fluorescence signals were acquired using a two-dimensional InGaAs array (Suzhou Optics) and thus fluorescence images were collected. Excitation light was provided by the 808 nm diode laser. The emitted light from animals was filtered through a 1000-nm long-pass filter for NIR-II imaging coupled with an InGaAs camera. The exposure time for all images was 250 ms.

**Ex vivo biodistribution analysis**. Ex vivo fluorescence imaging of organs and tissues were used Suzhou Optics NIR-II fluorescence imaging system with an InGaAs camera at a power density of ~80 mW/cm² at 808 nm laser diode irradiation. Forty-eight hours after injection of H4-PEG-PT for NIR-II imaging, 143B xenograft mice were sacrificed, the NIR-II images of major organs were collected.

**Histological analysis**. Major organs and tumors were taken from the tumor-bearing mice and fixed with EDTA/formalin solution. After embedment and section, tissue samples were further stained with hematoxylin and eosin and subsequently imaged using a NIKON Eclipse ci microscope with ×40 and ×200 magnification.

**Calculation of the photothermal conversion efficiency**. The photothermal conversion efficiency of cocrystal was determined according to previous methods.

$$\eta = \frac{hS\Delta T_{max} - Q_s}{I(1 - 10^{-A_{808}})} \quad (2)$$

$$\tau = \frac{\sum_i m_i C_i}{hs} \quad (3)$$

In order to get the $h*S$, a dimensionless driving force temperature, $\theta$ is introduced as follows:

$$\Theta = \frac{T - T_{sur}}{T_{max} - T_{sur}} \quad (4)$$

Thus,

$$\frac{d\theta}{dt} = \frac{1}{\tau} \frac{Q_s}{hS\Delta T_{max}} \times (-) \frac{\theta}{\tau} \quad (5)$$

When the laser is off, $Q_s = 0$, therefore $\frac{d\theta}{dt} = -\frac{\theta}{\tau}$,

$$t = -\tau \ln\theta \quad (6)$$

Where $I$ is the laser power (2.0 W cm$^{-2}$) and A808 is the absorbance of the samples at the wavelength of 808 nm (0.13). $T_{max} = 67.5$ °C, $T_{sur} = 23.8$ °C.

$Q_s$ is the heat associated with light absorption by the solvent. The variable $\tau$ is the sample-system time constant, and $m$ and $C$ are the mass and heat capacity of the deionized water used as the solvent. According to Eqs. (2) and (3), the $\eta$ value of the H4-PEG-PT was determined to be ~18%.

**In vitro photothermal therapy of 143B cells with H4-PEG-PT**. For the evaluation of photothermal therapeutic efficiency, MTT assay was used to measure the

cell viability after laser irradiation. 143B cells were seeded in a 96-well plate at a density of 5000 or 6000 cells per well for 12 h in a standard cell culture atmosphere. The cells were added with H4-PEG-PT at its concentration of 8, 16, 32, and 64 μM. After incubation for 6 h, a part of cells in the wells was irradiated with an 808 nm NIR laser (1.8 and 2.0 W cm$^{-2}$) for 0 and 4 min. Afterward, the cell viability of 143B was measured using a standard MTT assay.

**In vivo photothermal therapy**. For photothermal therapy, the nude mice bearing subcutaneous 143B tumors were randomly divided into three groups ($n = 4$ per group): (a) PBS injection group with 808 nm laser, (b) H4-PEG-PT injection group (200 μL, 400 μg per mouse), (c) H4-PEG-PT injection group (200 μL, 400 μg per mouse) with 808 nm laser. For groups (a) and (b), after 12 h post-injection, right tumors were irradiated by the 808 nm laser at a power density of 1.5 W cm$^{-2}$ (CAUTION: this power density may cause skin hurt!) for 8 min. During the treatment, the tumor lengths and widths were measured by a digital caliper every 2 days for 2 weeks. The tumor volume (mm³) was calculated by the formula: tumor volume = length × (width)²/2.

**Ethics statement**. All animal studies were performed in accordance with the Guidelines for the Care and Use of Laboratory Animals of the Chinese Animal Welfare Committee and approved by The Institutional Animal Care and Use Committee (IACUC), Wuhan University Center for Animal Experiment, Wuhan, China.

**Data analysis**. All statistical data were expressed as means ± standard deviation. Statistical differences were analyzed with Student's t test to compare the differences between two groups. Quantum chemistry calculation was performed using Gaussian 09 program (Revision D.09) software, Image J (1.51j8) was used to analyze the NIR-II images. Origin 9 was used to analyze the spectrum images. All statistical analyses were performed by Graphpad Prism 8.0 and SPSS 20.0.

**Reporting summary**. Further information on research design is available in the Nature Research Reporting Summary linked to this article.

## Data availability

The authors declare that all data needed to evaluate the conclusion of this work are presented in the paper and the Supplementary Information. Additional data related to this paper may be requested from the authors. Source data are provided with this paper.

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

## Acknowledgements

The work was supported by the National Key R&D Program of China (2020YFA0908800), NSFC (81773674, 81573383), Shenzhen Science and Technology Research Grant (JCYJ20190808152019182), Hubei Province Scientific and Technical Innovation Key Project, National Natural Science Foundation of Hubei Province (2017CFA024, 2017CFB711), the Applied Basic Research Program of Wuhan Municipal Bureau of Science and Technology (2019020701011429), Tibet Autonomous Region Science and Technology Plan Project Key Project (XZ201901-GB-11), the Local Development Funds of Science and Technology Department of Tibet (XZ202001YD0028C), Project First-Class Disciplines Development Supported by Chengdu University of Traditional Chinese Medicine (CZYJC1903), Health Commission of Hubei Province Scientific Research Project (WJ2019M177, WJ2019M178), the China Scholarship Council, and the Fundamental Research Funds for the Central Universities.

## Author contributions

X.H. and Y.X. designed the research. Y.X., H.Z., X.Z., A.L., L.T., and W.Z. performed research. X.H., Y.X., H.Z., X.Z., A.L., L.T., W.Z., X.M., W.H., Q.F., H.D., L.D., Y.L., and Z.D. contributed analytic tools and data analyses. X.H., H.Z., Y.X., and X.Z. wrote the paper.

## Competing interests

The authors declare no competing interests.
