## [Peer Review File · Nature Communications]

Reviewers' Comments:

Reviewer #1:

Remarks to the Author:

This is an interesting paper that describes a series of new D-A type thiopyrylium-based probes, with tunable emission wavelengths from 600 to 1100 nm. The probes modified with PEG and osteosarcoma-targeting oligopeptide showed preferential accumulation in cancer cells and were well retained in mitochondria through their delocalized lipophilic cationic properties even after fixation and permeabilization. The highlighted probe H4-PEG-PT displayed frequency upconversion luminescence (FUCL), NIR II emission and photothermal property. The H4-PEG-PT was further evaluated to exhibit in vitro good tumor uptake by upconversion-mitochondria-targeted cell imaging and in vivo NIR-II osteosarcoma imaging and excellent mice tumor PTT.

1, In the abstract, the investigation of cellular uptake should be in vitro experiments.

2, In figure 1B, the n in the chemical structure should be defined.

3, The calculated optimized structures in figure 2 may be obtained by the Gauss software. The detailed information of the software should be given, in case the software copyright infringements.

4, Why disruptively dropped emissions in the curves were seen in figure 4F. Please explain.

5, What are the nano sizes of used chemical structures in aqueous solution. The data should be provided. How about the stability?

6, In the in vivo imaging of osteosarcoma mice, by blocking, the nanoparticles can barely accumulate in tumor. Usually, the nanoparticles have EPR effect which can go into the tumor with passive targeting. Please explain this phenomenon. In Figure S5, how many mice were used to measure the relative fluorescence intensity? The statistical analysis should be added.

7, In figure 8b, the data of curves were repeatedly used in the first and last loop. Please double check.

8, In figure 8J, the authors claim the highest fluorescence of Cyto c was observed in the group treated with H4-PEG-PT and 808 nm laser irradiation. Please compare the intensity to support your claim.

Reviewer #2:

Remarks to the Author:

The manuscript entitled "Upconversion NIR-II Fluorophores for Mitochondria-Targeted Cancer Imaging and Photothermal Therapy" describes a novel NIR II dye, able to selectively accumulate to cancer cell mitochondria and stimulated mitochondria-dependent cell death after irradiation. The novel compound is of significant interest as potential marker for cancer imaging or for the development of anti-cancer phototherapy.

This observation it is sufficiently novel to be of interest for the readership of Nature Communications, therefore this reviewer supports its consideration by the journal.

At the present stage, the study appears immature for the publication and a revision of some aspects is recommended. Will follow a list of concerns that authors should address before manuscript publication:

Four major concern are raised by this referee:

a. The mitochondrial localization. Mitochondrial localization is predicted by the meaning of the cationic moiety and confirmed by confocal imaging of the novel compound and the mitochondrial marker Mitotracker Red or Green. Presented images are at too poor resolution to support the statement and the mitotracker staining have poor quality. Indeed, it seems to have stained members of the endomembrane compartments (this often happen with mitotracker dyes). This rise doubts in this referee about the selective localization to the mitochondrial compartment. Authors should perform colocalization measurement with more reliable mitochondrial markers as well as markers of the endomembrane system (at least endoplasmic reticulum). A preferable strategy would be the use of targeted GFP/RFP, but immunostaining could also work (when specific). Colocalization should be quantified using dedicated indexes (e.g. Pearson or Mander's coefficient).

Also, this referee wonder if the localization to mitochondria is actually driven by the cationic moiety. Is the uptake of NIR II dye dependent on mitochondrial membrane potential (could be blocked by the use of uncouplers)? This would provide useful insights for the future developments of its use.

b. Characterization of mitochondrial physiology. Mitochondrial physiology is surprisingly not investigated at all. It is known that multiple mitochondrial dye can affect some mitochondrial enzymatic activities. A basic description of mitochondrial physiology should be then included with and without NIR excitation. The referee would recommend running respiration measurements (e.g. Seahorse Mitostress test), ATP production, mitochondrial membrane potential and mitochondrial morphology analysis.

Also ROS should be better reported, the presented images of MitoSOX are completely black, this referee cannot understand if there is no ROS production (which is unlikely also at baseline) or if there have been issues with the assay. Measurements of H₂O₂ (e.g. by the use of the HyPer reporter) might also be informative.

c. Description of mitochondrial-dependent cell death. Authors implies that the mitochondrial localization is sufficient to justify mitochondrial-dependent cell death and reported cytochrome C staining together with Caspases 3 and 9 conversion from the zymogen. Unfortunately, the Cytochondrme C staining is not conclusive (this referee cannot detect significant differences in the representative images) and Caspases activation is not always a marker of cell death. A better description of this process should be reported.

As for early stage of cell death induction, authors should better describe marker of mitochondrial outer membrane permeabilization (MOMP) which includes release of other mitochondrial proteins (i.e. SMAC/DIABLO, EndoG, AIF) and relocation of Bax and Bak. Also, Authors should evaluate the hypothesis that mitochondrial permeability transition MPT occurs. Several assays are available in this term, nonetheless, Calcein-Cobalt and TMRM assays are the most reliable.

For effector phase, activity of Caspases should be confirmed by enzymatic kit (there are many options available) or by detection of the cleaved targets of Caspases (e.g. Lamin A/C or PARP).

Finally, a late stage marker would also be required, especially an annexin V/PI test would confirm engagement of apoptosis VS necrosis or mixed paths.

d. Selectivity for cancer cells. Authors states that the investigated compound is selectively up taken by cancer cells. Nonetheless only one type of cancer cell has been investigated in vivo. This referee recommends comparing uptake capacity and sensitivity to irradiation (a simple readout of vitality or cell death would work) between different transformed and primary cell cultures. After that at least another type of cancer cells should be tested in vivo.

Reply to Reviewer 1:

We appreciate the reviewer's insightful comments.

This is an interesting paper that describes a series of new D-A type thiopyrylium-based probes, with tunable emission wavelengths from 600 to 1100 nm. The probes modified with PEG and osteosarcoma-targeting oligopeptide showed preferential accumulation in cancer cells and were well retained in mitochondria through their delocalized lipophilic cationic properties even after fixation and permeabilization. The highlighted probe **H4-PEG-PT** displayed frequency upconversion luminescence (FUCL), NIR II emission and photothermal property. The **H4-PEG-PT** was further evaluated to exhibit *in vitro* good tumor uptake by upconversion-mitochondria-targeted cell imaging and *in vivo* NIR-II osteosarcoma imaging and excellent mice tumor PTT.

1. In the abstract, the investigation of cellular uptake should be *in vitro* experiments.

Reply to the reviewer: Thanks for this meaningful question. The cellular uptake of 143B cells treated with **H4-PEG-PT** was illustrated in **Supplementary Fig. 4**. HepG2 cells were included as the control.

The following information has been updated in the revised SI on page 9:

“The cellular uptake of 143B and HepG2 cells treated with H4-PEG-PT

Human bone osteosarcoma 143B cells and human hepatocellular carcinoma HepG2 cells were dispersed in 20 mm confocal dishes, respectively, and cultured for 12 h at a density of 1×10^5 cells/well. Then, 200 μ L of **H4-PEG-PT** (64 μ M) was introduced into the confocal dishes. After 7 h incubation, the cells were washed twice with PBS, stained with Hoechst staining kit in PBS for 20 min. The tumor cell samples were immediately examined by an inverted Zeiss Axiovert 200 fluorescence microscopy at a 20 \times objective, using a high-pressure mercury lamp as an excitation source.”

The following data have been updated in the revised SI:

Supplementary Fig 4. The *in vitro* cellular uptake of **H4-PEG-PT** was investigated against 143B cells. Scale bar = 10 μm .

2. In **Figure 1B**, the n in the chemical structure should be defined.

Reply to the reviewer: We thank the reviewer for pointing this out. $n = 0, 1$ in the chemical structure of **Fig. 1b** has been defined accordingly. The modified figure was shown as below.

3. The calculated optimized structures in **Figure 2** may be obtained by the Gauss software. The detailed information of the software should be given, in case the software

copyright infringements.

Reply to the reviewer: Thanks for indicating this issue. Density functional theory (DFT) calculations were carried out by the Lee Yang Parr's correlation functional (B3LYP) using 6-31G(d) basis sets, Gaussian 09, Revision D.09..

The following information has been updated in the revised manuscript in **Fig. 2**:

“(b) Calculated optimized ground state (S_0) geometries of the molecules at the B3LYP/6-31G (d) level (Gaussian 09, Revision D.09).”

4. Why disruptively dropped emissions in the curves were seen in **Figure 4F**. Please explain.

Reply to the reviewer: Thanks for this critical question. Two emission bands were observed at ~ 1100 and 1200 nm in DMSO solvent. This phenomenon may be attributed to the solvent absorption and the twisted intramolecular charge transfer (TICT) effect (*Nanoscale* **2015**, 7, 133; *Adv. Mater.* **2017**, 1605497). Dual-peak emission spectra were also observed in D-A-D type molecules with TICT effects in the visible range (*Chem. Rev.* **2003**, 103, 3899).

5. What are the nano sizes of used chemical structures in aqueous solution. The data should be provided. How about the stability?

Reply to the reviewer: We thank you for raising this critical issue. **H4-PEG-PT** readily formed supramolecular assemblies in water with the average length of 180.0 ± 13 nm and the average width of 48 ± 15 nm as determined by transmission electron microscopy (TEM, **Fig. 5i**). Besides, the stability of **H4-PEG-PT** was measured by continuous laser irradiation in different media as shown in **Fig. 5e**.

Fig. 5. (e) Compared stability of **H4-PEG-PT** vs. ICG in different media under continuous laser irradiation.

The following text and data were added in the revised manuscript on page 10:

“**H4-PEG-PT** readily formed supramolecular assemblies in water with the average length

of 180.0 ± 13 nm and the average width of 48 ± 15 nm as determined by transmission electron microscopy (TEM, **Fig. 5i**).”

Fig. 5. (j) TEM image of **H4-PEG-PT** (scale bar: 500 μ m).

6. In the *in vivo* imaging of osteosarcoma mice, by blocking, the nanoparticles can barely accumulate in tumor. Usually, the nanoparticles have EPR effect which can go into the tumor with passive targeting. Please explain this phenomenon. In **Supplementary Fig. 8**, how many mice were used to measure the relative fluorescence intensity? The statistical analysis should be added.

Reply to the reviewer: Thanks for this critical question. The oligopeptide PPSHTPT (**PT**) was first designed to mimic the properties of the natural protein osteocalcin *in vivo* and possessed a high affinity and specificity for osteosarcoma cell lines (e.g.143B cells). **H4-PEG-PT** with the target peptide **PT** was demonstrated to preferentially accumulate in osteosarcoma cells *in vitro* by active targeting. Much higher fluorescence was observed in the **non-blocking** experiment *in vivo* (left tube, **Fig. 5d**) while negligible fluorescence was detected in **the blocking** experiment (right tube, **Fig. 5d**). Although **H4-PEG-PT** readily formed supramolecular assemblies in water, the active targeting ability of **H4-PEG-PT** was found to be dominated than EPR effects according to the blocking experiments. EPR effect has paved the way for the passive targeting of tumors for nanoparticles, however, sometimes the EPR effect of nanoparticle may be limited, as the rate of leakage from the vessels is slow and the NPs can either be excreted or metabolized during the time it takes for the accumulation to tumors. Several factors could also influence the EPR effect including the nature of both the vascular bed and surrounding stroma, tumor size, type, and location, and extent of macrophage tumor infiltration and the activity of the mononuclear phagocytic system etc.

In **Supplementary Fig. 8**, three mice were chosen to measure the relative fluorescent intensity. The relative information was updated in the **Supplementary Fig. 8**.

Supplementary Fig. 8. The biodistribution of **H4-PEG-PT** in orthotopic tumor mice at 48 h under an 808 nm excitation (1000LP, 3.5W and 200 ms) and the relative fluorescence intensity analysis of different organs (n = 3).

7. In **Figure 8b**, the data of curves were repeatedly used in the first and last loop. Please double check.

Reply to the reviewer: Thank you very much for your valuable suggestion. We have double-checked the curves as shown in **Fig. 8b**. The first and last loops were very similar, indicating excellent thermal and photo-stability of **H4-PEG-PT** over several ON/OFF cycles involving irradiation with an 808 nm laser (2.0 W/cm^2) for 80 min followed by passive cooling. The original data have been attached in the uploading file (named as "source data").

8. In **Figure 8J**, the authors claim the highest fluorescence of Cyto c was observed in the group treated with **H4-PEG-PT** and 808 nm laser irradiation. Please compare the intensity to support your claim.

Reply to the reviewer: Thanks for your constructive suggestion. The fluorescent intensity of Cyto c was quantified as below. We can clearly find that under 808 nm laser irradiation, the highest fluorescence of Cyto c was observed in the group treated with **H4-PEG-PT**. These results have demonstrated that photothermal therapy of **H4-PEG-PT** against 143B cells led to Cyto c release and cell apoptosis.

The following data were added to the revised SI on page 18:

Supplementary Fig. 11. Quantitative analysis of Cyto c was measured in different groups by the photothermal therapy.

Reply to Reviewer 2:

The manuscript entitled “Upconversion NIR-II Fluorophores for Mitochondria-Targeted Cancer Imaging and Photothermal Therapy” describes a novel NIR II dye, able to selectively accumulate to cancer cell mitochondria and stimulated mitochondria-dependent cell death after irradiation.

The novel compound is of significant interest as potential marker for cancer imaging or for the development of anti-cancer phototherapy.

This observation it is sufficiently novel to be of interest for the readership of *Nature Communications*, therefore this reviewer supports its consideration by the journal.

At the present stage, the study appears immature for the publication and a revision of some aspects is recommended. Will follow a list of concerns that authors should address before manuscript publication:

Reply to the reviewer: Thank you very much for your positive comments and insightful suggestions, which highly improve the quality of our manuscript.

Four major concern are raised by this referee:

a. The mitochondrial localization. Mitochondrial localization is predicted by the meaning of the cationic moiety and confirmed by confocal imaging of the novel compound and the mitochondrial marker Mitotracker Red or Green. Presented images are at too poor resolution to support the statement and the mitotracker staining have poor quality. Indeed, it seems to have stained members of the endomembrane compartments (this often happen with mitotracker dyes). This rise doubts in this referee about the selective localization to the mitochondrial compartment. Authors should perform colocalization measurement with more reliable mitochondrial markers as well as markers of the endomembrane system (at least endoplasmic reticulum). A preferable strategy would be the use of targeted GFP/RFP, but immunostaining could also work (when specific) Colocalization should be quantified using dedicated indexes (e.g. Pearson or Mander's coefficient).

Also, this referee wonder if the localization to mitochondria is actually driven by the cationic moiety. Is the uptake of NIR II dye dependent on mitochondrial membrane potential (could be blocked by the use of uncouplers)? This would provide useful insights for the future developments of its use.

Reply to the reviewer: Thanks for this critical question, which is vital to improve the quality of our manuscript. To improve the resolution, high quality of the mitotracker staining was also conducted and shown in the following.

Fig. 6(a) Fluorescent images of 143B cells incubated with **3j-PEG** (10 μ M) for 8 h (left); with Mito-tracker red (middle) and merge images of them (right). λ_{em} : 590-650 nm, λ_{ex} : 594 nm (Mito-tracker red); λ_{em} : 490 nm-540 nm, λ_{ex} : 488 nm (**3j-PEG**)

Mitochondria-GFP provides an easy way to label mitochondria with green fluorescent protein (GFP) in live cells. As suggested by the referee, more reliable colocalization measurement experiments using GFP was performed.

The following data were added to the revised manuscript on page 12:

Fig. 6 (f) Fluorescent images of 143B cells incubated with Mitochondria-GFP (left), **H4-PEG-PT** (middle) and merge images of them (right).

We also have updated this in the revised manuscript (Page 12) and **Fig. 6:**

Fig. 6. Mitochondrial localization. **(a)** Fluorescent images of 143B cells incubated with **3j-PEG** (10 μ M) for 8 h (left); with Mito-tracker red (middle) and merge images of them (right). λ_{em} : 590-650 nm, λ_{ex} : 594 nm (Mito-tracker red); λ_{em} : 490 nm-540 nm, λ_{ex} : 488 nm (**3j-PEG**). **(b)** Merge images of **3j-PEG** and Mito-tracker red in 3D fluorescence images of mitochondrial localization. Fluorescent images of 143B cells incubated with Mito-tracker green (left); with (c) **3k-PEG** (1 μ M) or (d) **H4-PEG** (1 μ M) or (e) **H4-PEG-PT** (1 μ M) for 6 h (middle) and merge images them (right).. λ_{em} : 495-526 nm, λ_{ex} : 488 nm (Mito-tracker green); λ_{em} : 610 nm-660 nm, λ_{ex} : 514 nm (**3k-PEG**, **H4-PEG** and **H4-PEG-PT**). (f) Fluorescent images of 143B cells incubated with Mitochondria-GFP (left), **H4-PEG-PT** (middle) and merge images of them (right). λ_{em} : 495-526 nm, λ_{ex} : 488 nm (Mitochondria-GFP); λ_{em} : 610 nm-660 nm, λ_{ex} : 514 nm (**H4-PEG-PT**). (scale bar: 20 μ m).

The Pearson correlation coefficient (PC) and the Mander's overlap coefficient (MOC) are widely used to quantify the degree of colocalization between fluorophores. Herein, the results of fluorescence colocalization studies are represented by the use of PC as a statistic for quantifying colocalization. As a consequence, while measurement of the PC for **Fig. 6a**, **Fig. 6c-e** indicates a reasonably strong correlation (0.76, 0.84, 0.82, and 0.88 in the indicated ROIs). Moreover, mitotracker dyes have been utilized for colocalization of mitochondria in the literatures (*Cancer Letters*, **2009**, 277, 64–71; *Chem. Sci.*, **2019**, 10, 1994–2000; *J Am Chem Soc.*, **2017**, 139, 9972–9978; *J. Am. Chem. Soc.*, **2016**, 138, 12368–12374). Thus, mitotracker green and red are selected for the colocalization of mitochondria in the manuscript.

The following data were added to the revised SI on page 15

Supplementary Fig. 13. (A) Colocalization scatterplots of **3j-PEG** and mito-tracker red, PC = 0.76. (B) Colocalization scatterplots of **3k-PEG** and mito-tracker green, PC = 0.84. (C) Colocalization scatterplots of **H4-PEG** and mito-tracker green, PC = 0.82. (D) Colocalization scatterplots of **H4-PEG-PT** and mito-tracker green. PC = 0.88.

Several lipophilic cationic dyes distribute electrophoretically into the mitochondrial matrix in response to the electric potential across the inner mitochondrial membrane. The accumulation takes place as a consequence of their charge and of their solubility in both the inner membrane lipids and the matrix aqueous space. For the above reason, these dyes have been extensively employed to measure the mitochondrial electric potential (*Biotechniques* **2011**, 50, 98; *Biochimica et Biophysica Acta (BBA)-Bioenergetics*, **2003**, 1606, 137; *ChemBioChem* **2009**, 10, 1939; *Chem. Sci.*, **2019**, 10, 1994–2000; *Angew. Chem. Int. Ed.* **2018**, 57, 16506-16510). Thus, experiments to confirm whether the NIR dye's uptake is dependent on mitochondrial membrane potential was carried out by the uncoupler assay. Carbonyl cyanide *p*-(trifluoromethoxy)phenylhydrazone (FCCP) is a mitochondrial uncoupler that causes rapid acidification of the mitochondria and dysfunction of ATP synthase and reduces the mitochondrial membrane potential. Mitotracker deep red was selected as control. Both of the two probes showed considerable reduction in the fluorescence intensity upon FCCP induced MMP depolarization. These results suggested stronger evidence that the synthesized fluorophore localized in mitochondria and its cellular uptake is dependent on the mitochondrial membrane potential.

Fluorescent images of 143B cells (only incubated with **H4-PEG-PT** or pretreated w/o FCCP (40 μ M) for 1 h before incubating with **H4-PEG-PT** (2 μ M) for 6 h, Mitotracker deep red was selected as control). Negligible fluorescence was observed in the FCCP treated groups.

b. Characterization of mitochondrial physiology. Mitochondrial physiology is surprisingly not investigated at all. It is known that multiple mitochondrial dye can affect some mitochondrial enzymatic activities. A basic description of mitochondrial physiology should be then included with and without NIR excitation. The referee would recommend running respiration measurements (e.g. Seahorse Mitostress test), ATP production, mitochondrial membrane potential and mitochondrial morphology analysis. Also ROS should be better reported, the presented images of MitoSOX are completely black, this referee cannot understand if there is no ROS production (which is unlikely also at baseline) or if there have been issues with the assay. Measurements of H₂O₂ (e.g. by the use of the HyPer reporter) might also be informative.

Reply to the reviewer: Thanks for your insightful suggestion.

1) The ATP levels were measured using the ATP determination kit. (No. A22066) based on luminescence from luciferase and luciferin. As shown below, the ATP levels of **H4-PEG-PT** with NIR irradiation (808 nm) were ~ 5.6% compared with the control group. Neither **H4-PEG-PT** nor NIR irradiation alone had any obvious changes compared with the control. At the same time, 143B tumor cells treated with carbonyl cyanide *m*-chlorophenylhydrazone (CCCP), a mitochondrial uncoupling agent showed relative lower ATP level (~ 78%) compared with the control, **H4-PEG-PT** or NIR irradiation (808 nm), indicating the mitochondrial physiology dysfunction of 143B cells.

The following data were added to the revised manuscript on page 18:

Fig. 8 (d) The cellular ATP production of 143B cells treated with **H4-PEG-PT** w/o 808 nm laser irradiation, CCCP was the positive control. The ATP level of control was 100%. (** p < 0.01, *** p < 0.001).

The following sentences were updated in the revised SI on page 9.

“Detection of ATP level.

143B cells (6×10^3) were incubated in 96 wells plate at 37 °C for 12 h. Then, cells were treated with 64 μM **H4-PEG-PT** or 40 μM CCCP for 6 h and irradiated by an 808 nm laser for 3 min each well. After irradiation, ATP level was immediately determined by an ATP determination kit (cat. no. A22066) from Thermo Fisher Scientific with 1% Triton X 100 water solution for cell permeabilization. After 10 min incubation, the luminescence intensity was then detected using a microplate reader (TECAN, Infinite M200 Pro).”

JC-1 dye can be used as an indicator of mitochondrial membrane potential in a variety of cell types. It exhibits potential-dependent accumulation in mitochondria, indicated by a fluorescence emission shift from green (~ 529 nm) to red (~ 590 nm). To further characterize mitochondrial physiology of 143B tumor cells treated with **H4-PEG-PT** under an 808 nm laser irradiation, the mitochondrial membrane potential was measured using 5,5',6,6'-tetrachloro-1,1',3,3'-tetraethyl-imidacarbocyanine iodide (JC-1) assay kit. In **Fig. 8e**, the ratios of J-aggregates to monomers of 808 nm laser irradiation or **H4-PEG-PT** treatment were 96%, and 109%, respectively, which is close to that of control cells (100%), indicated good biocompatibility of laser irradiation or **H4-PEG-PT** treatment. However, the ratios of J-aggregates to monomers of 143B tumor cells treated with **H4-PEG-PT** under NIR irradiation or CCCP were decreased to 25% and 63%, respectively. The apparent change of mitochondrial membrane potential has been shown by **H4-PEG-PT** treatment with NIR irradiation, indicating mitochondrial dysfunction of 143B tumor cells during the photothermal therapy using **H4-PEG-PT**.

The following data were added to the revised manuscript on page 18:

Fig. 8 (e) Mitochondrial membrane potential quantified by JC-1 fluorescence assay of 143B cells treated with **H4-PEG-PT** w/o 808 nm laser irradiation, CCCP was the positive control.

The following sentences were added to the revised SI on page 9:

“Detection of mitochondrial membrane potential by JC-1 assay.

143B cells (6×10^3) were incubated in 96 wells plate for 12 h. Then, cells were treated with **H4-PEG-PT** (64 μM), or CCCP (40 μM) for 6 h and irradiated by an 808 nm laser for 3 min. After irradiation, JC-1 (1 μM) was immediately added to each well for 30 min incubation at 37 °C. Fluorescence intensity was then detected using a microplate reader (TECAN, Infinite M200 Pro) with excitation at 485 nm (green) or 535 nm (red) and emissions at 535 nm (green) or 595 nm (red). The ratio of red fluorescence intensity (J-aggregates) to green fluorescence intensity (monomers) was used to quantify the mitochondrial membrane potential change of **H4-PEG-PT** with laser irradiation, the ratio of the control was 100%.”

Furthermore, calcein-AM cobalt assay was performed to identify mitochondrial permeability transition (MPT) of **H4-PEG-PT** with NIR irradiation in 143B cells. Hydrogen peroxide is used as a positive agent. As shown in **Fig. 8f**, the calcein fluorescence intensity of 143B cells treated with 808 laser irradiation or **H4-PEG-PT** was similar to that of the control. After adding H_2O_2 , the fluorescence intensity was sharply decreased due to the occurrence of mitochondrial permeability transition in 143B cells. Meanwhile, 143B cells incubated with **H4-PEG-PT** under NIR irradiation exhibited much lower fluorescence signal than that of H_2O_2 -induced group.

The following data were added to the revised manuscript on page 18:

Fig. 8 (f) Mitochondrial permeability transition (MPT) study of **H4-PEG-PT** after 808 laser irradiation using calcein-AM cobalt assay. H_2O_2 treatment is the positive control.

The following sentences were added to the revised SI on page 10:

“Detection of mitochondrial permeability transition

143B cells (6×10^3) were incubated in 96 wells plate for 12 h. Then, 100 μ L of **H4-PEG-PT** (64 μ M) in fresh DMEM medium was introduced into the 96 wells plate. After an incubation time of 6 h, cells were irradiated by an 808 nm laser for 3 min. After that, cells were loaded with 1 μ M of calcein-acetoxymethyl ester (AM) in Hank’s buffer at 37 °C for 10 min and treated with 2 mM cobalt chloride or 2 mM cobalt chloride plus 10 mM hydrogen peroxide Hank’s buffer solution for 10 min incubation at 37 °C. The fluorescence intensity of cell samples was measured using a microplate reader (TECAN, Infinite M200 Pro) with 488 nm excitation and 535 nm emission filters.”

The following sentences were added to the revised manuscript on page 15:

“To explore the plausible mechanisms of **H4-PEG-PT** under NIR laser irradiation against 143B cells, its effects on mitochondrial function and apoptosis were investigated (**Fig. 8d-f**). Mitochondrial functioning was first assessed by determining the levels of ATP in 143B cells (**Fig. 8d**). Pretreatment with **H4-PEG-PT** or 808 nm irradiation, the levels of ATP were similar to those in control cultures. **H4-PEG-PT** under NIR irradiation treatment significantly decreased 143B cellular ATP generation to ~ 5.6% (**Fig. 8d**). JC-1 assay was further used to analyze mitochondrial membrane potential ($\Delta\Psi_m$). The ratio of J-aggregates to monomers of **H4-PEG-PT** (60 μ M) or NIR irradiation treatments for up to 3 min had no significant decrease. Treatment with **H4-PEG-PT** (60 μ M) under NIR irradiation or carbonyl cyanide *m*-chlorophenylhydrazone (CCCP) reduced the ratio of J-aggregates to monomers of 143B tumor cells to 25% and 63%, respectively (**Fig. 8e**). The change in the permeability of mitochondrial membranes is a significant event in the apoptosis process, which is controlled by and mitochondrial permeability transition (MPT). In our experiment, calcein-AM cobalt assay was used to identify MPT. The experimental results showed that, after 143B cells were treated with **H4-PEG-PT** under an 808 nm laser irradiation for 3 min, the calcein fluorescence intensity of 143B cells was much lower fluorescence signal than that of H₂O₂-induced group (**Fig. 8f**). These results revealed that **H4-PEG-PT** under NIR irradiation would cause serious mitochondrial physiology dysfunction, while **H4-PEG-PT** or NIR irradiation treatment alone could not damage cells.”

2) The light red fluorescence was observed in the group of “808 nm laser + MitoSOX” and “808 nm laser + MitoSOX + NAC”, which could demonstrate that negligible ROX was generated by **H4-PEG-PT** during the 808 nm laser irradiation. These results were consistent with that of DPBF (ROS indicator) *in vitro* as shown in **Fig. 8i-p**. Besides, MitoSOX, a mitochondrial ROS sensor, is the most common dye to detect mitochondrial ROS levels, if the probes have negligible ability to generate ROS, then the images of MitoSOX are almost black. Similar references have reported the results (*J Am Chem Soc.* **2017**, 139, 9972–9978)

A higher resolution picture of **Fig. 8r** has been updated in the manuscript on page 19:

Fig. 8 (r) Confocal fluorescence microscopic images of 143B cells incubated with **H4-PEG-PT** (64 μ M) and treated MitoSOX (1.0 μ M) according to the treatment variables, including w/o 808 nm NIR irradiation (1 W/cm²) for 5 min. Scale bar: 10 μ m.

DCFH-DA was also selected to measure H₂O₂ according to the previous literatures (*Adv. Funct. Mater.* **2015**, *25*, 7280–7290; *Biomaterials* **2014**, *35* 1954-1966) since HyperRed (DOI: 10.1038/ncomms6222) was not commercial unavailable in Sigma-Aldrich, Aladdin and some other chemical reagent companies (*Nat. Commun.* **2014**, *5*, 5222). As shown in the figures below, no significant fluorescence was observed in the two groups. All the results showed that **H4-PEG-PT** could generate negligible ROS during the 808 nm laser irradiation.

The following sentences were added to the revised SI on page 10:

“Dichloro-dihydro-fluorescein diacetate (DCFH-DA) assay.

143 B cells (1×10^4 cells/well) were seeded into a 96-well plate and incubated with 30 μ M **H4-PEG-PT** for 6 h and 10 μ M DCFH-DA (Ex/Em = 504/529 nm) for 30 min. The cells were then exposed to laser irradiation (2 W/cm², 808 nm) for 3 min prior imaging. The tumor cell samples were immediately examined by an inverted Zeiss Axiovert 200 fluorescence microscopy at a 10 \times objective, using a high-pressure mercury lamp as excitation source.”

The following sentences were added to the revised manuscript on page 16:

“NIR light-triggered ROS generation of **H4-PEG-PT** under 808 nm laser irradiation was then evaluated by using 1,3-diphenylisobenzofuran (DPBF), dichloro-dihydro-fluorescein diacetate (DCFH-DA) and MitoSOX assay (**Fig. 8I-p**, **Fig 8r** and Supplementary **Fig. 12**)^{64,65}. No significant amount of ROS *in vitro* was observed in any group of **H4-PEG-PT** under 808 nm irradiation.”

The following data were added to the revised SI on page 18:

Supplementary Fig. 12. Intracellular ROS generation using DCFH-DA assay in 143B cells incubated with **H4-PEG-PT**, Scale bar: 100 μm .

c. Description of mitochondrial-dependent cell death. Authors implies that the mitochondrial localization is sufficient to justify mitochondrial-dependent cell death and reported cytochrome C staining together with Caspases 3 and 9 conversion from the zymogen. Unfortunately, the Cytochondrme C staining is not conclusive (this referee cannot detect significant differences in the representative images) and Caspases activation is not always a marker of cell death. A better description of this process should be reported.

As for early stage of cell death induction, authors should better describe marker of mitochondrial outer membrane permeabilization (MOMP) which includes release of other mitochondrial proteins (i.e. SMAC/DIABLO, EndoG, AIF) and relocation of Bax and Bak. Also, Authors should evaluate the hypothesis that mitochondrial permeability transition MPT occurs. Several assays are available in this term, nonetheless, Calcein-Cobalt and TMRM assays are the most reliable.

For effector phase, activity of Caspases should be confirmed by enzymatic kit (there are many options available) or by detection of the cleaved targets of Caspases (e.g. Lamin A/C or PARP).

Finally, a late stage marker would also be required, especially an annexin V/PI test would confirm engagement of apoptosis VS necrosis or mixed paths.

Reply to the reviewer: We highly appreciate your constructive suggestions.

1) Fluorescent intensity of Cyto c was quantified by software image J as below. We can clearly find that under 808 nm laser irradiation, the highest fluorescence of Cyto c was observed in the group treated with **H4-PEG-PT**. These results have demonstrated that photothermal therapy of **H4-PEG-PT** lead to Cyto c release and apoptosis.

The following data were added to the revised SI on page 18:

Supplementary Fig. 11. Quantitative analysis of Cyto c in different group by the

photothermal therapy.

2) Furthermore, calcein-AM cobalt assay was performed to identify mitochondrial permeability transition (MPT) of **H4-PEG-PT** with NIR irradiation in 143B cells. Hydrogen peroxide is used as a positive agent. As shown in **Fig. 8f**, the calcein fluorescence intensity of 143B cells treated with 808 laser irradiation or **H4-PEG-PT** was similar to that of the control. After adding H_2O_2 , the fluorescence intensity was sharply decreased due to the occurrence of mitochondrial permeability transition in 143B cells. Meanwhile, 143B cells incubated with **H4-PEG-PT** under NIR irradiation exhibited much lower fluorescence signal than that of H_2O_2 -induced group.

The following data were added to the revised manuscript on page 18:

Fig. 8 (f) Mitochondrial permeability transition (MPT) study of **H4-PEG-PT** after 808 nm laser irradiation using calcein-AM cobalt assay. H_2O_2 treatment is the positive control.

The following sentences were added to the revised SI on page 10:

“Detection of mitochondrial permeability transition

143B cells (6×10^3) were incubated in 96 wells plate for 12 h. Then, 100 μ L of **H4-PEG-PT** (64 μ M) in fresh DMEM medium was introduced into the 96 wells plate. After an incubation time of 6 h, cells were irradiated by an 808 nm laser for 3 min. After that, cells were loaded with 1 μ M of calcein-acetoxymethyl ester (AM) in Hank’s buffer at 37 °C for 10 min and treated with 2 mM cobalt chloride or 2 mM cobalt chloride plus 10 mM hydrogen peroxide Hank’s buffer solution for 10 min incubation at 37 °C. The fluorescence intensity of cell samples was measured using a microplate reader (TECAN, Infinite M200 Pro) with 488 nm excitation and 535 nm emission filters.”

The following sentences were added to the revised manuscript on page 15:

“To explore the plausible mechanisms of **H4-PEG-PT** under NIR laser irradiation against 143B cells, its effects on mitochondrial function and apoptosis were investigated (**Fig. 8d-f**). Mitochondrial functioning was first assessed by determining the levels of ATP in 143B cells (**Fig. 8d**). Pretreatment with **H4-PEG-PT** or 808 laser irradiation, the levels of ATP were similar to those in control cultures. **H4-PEG-PT** under NIR irradiation treatment significantly decreased 143B cellular ATP generation to ~ 5.6% (**Fig. 8d**). JC-1 assay was further used to analyze mitochondrial membrane potential ($\Delta\Psi_m$). The ratio of J-aggregates to monomers of **H4-PEG-PT** (60 μ M) or only NIR irradiation treatments for up to 3 min had no significant decrease. Treatment with **H4-PEG-PT** (60 μ M) under NIR irradiation or carbonyl cyanide *m*-chlorophenylhydrazone (CCCP) reduced the ratio of J-aggregates to monomers of 143B tumor cells to 25% and 63%, respectively (**Fig. 8e**). The change in the permeability of mitochondrial membranes is a significant event in the apoptosis process, which is controlled by and mitochondrial permeability transition (MPT). In our experiment, calcein-AM cobalt assay was used to identify MPT. The experimental results showed that, after 143B cells were treated with **H4-PEG-PT** under an 808 nm laser irradiation for 3 min, the calcein fluorescence intensity of 143B cells was much lower fluorescence signal than that of H₂O₂-induced group (**Fig. 8f**). These results revealed that **H4-PEG-PT** under NIR irradiation would cause serious mitochondrial physiology dysfunction, while **H4-PEG-PT** or NIR irradiation treatment alone could not damage cells.

3) Mitochondria plays a major role in regulating cell death, which occurs upon permeabilization of their membranes. Once mitochondrial membrane permeabilization (MOMP) occurs, cells die either by apoptosis or necrosis. PTT triggered mitochondrial damages with a strong release of mitochondrial proteins. When mitochondrial outer membrane permeabilization increases, AIF, EndoG and Smac/Diablo, which is usually on the inner mitochondrial membrane, are released from mitochondria into the cytosol and initiates apoptosis. Western blotting results have demonstrated that there was burst release of AIF, EndoG and Smac/DIABLO into the cytoplasm in **H4-PEG-PT** after NIR irradiation, which indicated the disruption of mitochondria and cellular metabolism. Meanwhile, the other three groups did not show elevated cytoplasm Cyto c and Smac/DIABLO level. It has also been shown that enhancement of the expression of proapoptotic Bax triggered mitochondrial dysfunction after irradiation and led to Cyto c release from the mitochondria (*Cancer Lett.* **2019**, 277, 2009, 64–71). PTT triggered mitochondrial damages have been reported with a strong release of Cyto c and Smac/Diablo (*Carbon*, **2016**, 97, 110-123). Therefore, mitochondrial damages could lead to Cyto c release and cell apoptosis because of photothermal therapy of **H4-PEG-PT**.

The following data were added to the revised manuscript on page 18:

Fig. 8(j) Representative western blot analysis of AIF, EndoG and Smac/Diablo in different groups.

The following sentences were added to the revised manuscript on page 16:

“Upon death stimuli, different apoptogenic factors such as apoptosis-inducing factor (AIF), Endonuclease G (EndoG) and second mitochondria-derived activator of caspase/direct IAP binding protein with low pI (Smac/Diablo) are released during the early stages of apoptosis. These three characteristic proteins of mitochondrial apoptosis were investigated by western blot. From **Fig. 8j**, the AIF and EndoG expression levels in the **H4-PEG-PT** group with NIR irradiation were significantly increased compared with that of the same group without NIR light irradiation, control group or NIR laser group. Western blotting results indicated that there was burst release of AIF, EndoG and Smac/DIABLO from mitochondria into the cytoplasm with **H4-PEG-PT** under irradiation, suggesting the disruption of mitochondria and cellular metabolism.”

4) Caspase-3/8/9 activities in 143B cells graphs were investigated by the Caspase multiplex activity assay Kit (Fluorometric) (ab219915). The increase of caspase-3/8/9 activities was also observed in the **H4-PEG-PT** group under NIR irradiation. Our findings further supported that cellular apoptosis of **H4-PEG-PT** was triggered by the intrinsic mitochondrial pathway.

The following sentences were added to the revised SI on page 10:

“Caspase-3/8/9 activities measured by caspase multiplex activity assay kit.

143 B cells were seeded on the same day at 1×10^5 cells/well in a clear bottom 96-well plate. DOX•HCl treatment is the positive control. Briefly, 143B cells (6×10^3) were seeded in a clear bottom 96 wells plate for 12 h. Then, 100 μ L of **H4-PEG-PT** (64 μ M) in fresh DMEM medium was introduced into the 96 wells plate. After an incubation time of 6 h, cells were irradiated by an 808 nm laser for 3 min. After that, cells incubated at 37 °C for 30 min, then Triple-caspase assay loading solution (100 μ L/well for Caspase 3, 8 and 9 together) was added to cells, followed by an incubation at RT for 30 min. The fluorescence intensity was measured at the indicated wavelength.”

The following sentences were added to the revised manuscript on page 16:

“Besides, the increase of caspase-3/8/9 activities was also observed in the H4-PEG-PT group under NIR irradiation by the caspase multiplex activity assay kit (Fig. 8g-i). Our findings further supported that cellular apoptosis of H4-PEG-PT was triggered by the intrinsic mitochondrial pathway.”

The following data were added to the revised manuscript on page 18:

Fig. 8(g)-(i). Caspase-3/8/9 activities measured by caspase multiplex activity assay kit in different groups.

5) In order to demonstrate the mechanism of fluorescence translocation phenomenon, FITC-annexin V and propidium iodide (PI) assay was used to identify early and late stage apoptotic versus necrosis. 143B cells were seeded in 96-well plates at a density of 1.2×10^4 cell/well. After treatment, the cells were harvested and re-suspended in binding buffer. Cells were stained with Annexin V-FITC and PI according to the manufacturer's instructions, followed by laser irradiation to induce cell death. Besides, we also used PI and Hoechst dual staining to evaluate the cellular apoptosis and necrosis. Normal cell nuclei are usually blue, while apoptotic cell nuclei are densely stained with bright blue.

The following sentences were added to the revised SI on page 10:

“Apoptosis or necrosis of 143B cells by Annexin V-FITC/PI and Hoechst/PI staining. 143B cells were seeded in 96-well plates at a density of 1.2×10^4 cell/well. After treatment,

the cells were harvested and re-suspended in binding buffer. Cells were stained with Annexin V-FITC and PI according to the manufacturer's instructions, followed by laser irradiation to induce cell death. Besides, we also used PI and Hoechst dual staining to evaluate the cellular apoptosis and necrosis. Normal cell nuclei are usually blue, while apoptotic cell nuclei are densely stained with bright blue."

The following data were added to the revised manuscript on page 19:

Fig. 8 (s) Apoptosis or necrosis of 143B cells determined by Annexin V-FITC/PI staining (scale bar: 50 μ m).

Fig. 8. (t) Apoptosis or necrosis of 143B cells determined by Hochest/PI staining (scale bar: 50 μ m).

The following sentences were added to the revised manuscript on page 17:

“FITC-annexin V and propidium iodide (PI) assay is usually used to identify early and late stage apoptotic versus necrosis. After 808 nm laser irradiation for 3 min, not only FITC signals were obviously emerged from the cell membrane but also PI signals were visible in the group **H4-PEG-PT** with laser irradiation. Then Hochest/PT staining assay was undertaken to examine the engagement of apoptosis VS necrosis or mixed paths. 143B cells treated with **H4-PEG-PT** under irradiation clearly demonstrated staining indicative of apoptosis (bright-blue particles). Because annexin V could stain early stage apoptotic cells while PI stains dead cells or late-stage apoptotic cells, and Hochest could stain the nuclei of apoptotic cells with a bright blue fluorescence, these two results showed that the cells treated with **H4-PEG-PT** under irradiation were engaged with both apoptosis and necrosis.”

d. Selectivity for cancer cells. Authors states that the investigated compound is selectively up taken by cancer cells. Nonetheless only one type of cancer cell has been investigated *in vivo*. This referee recommends comparing uptake capacity and sensitivity to irradiation (a simple readout of vitality or cell death would work) between different transformed and primary cell cultures. After that at least another type of cancer cells should be tested *in vivo*.

Reply to the reviewer: Thanks for this meaningful question. In order to confirm the selectivity of **H4-PEG-PT** for different cancer cells, the cellular uptake capacity of 143B and HepG2 cells to **H4-PEG-PT** was investigated by fluorescence microscopy. After incubation with **H4-PEG-PT** for 7 h, very strong fluorescence signals were observed located inside the 143B cells, revealing the accumulation of **H4-PEG-PT** into the osteosarcoma cells. However, hepatocellular carcinoma HepG2 cells showed very low level of fluorescence signals inside the cells at the same experimental condition. In addition, the binding ability of **H4-PEG-PT** to osteosarcoma was further confirmed by co-incubation of 143B cells with **H4-PEG-PT**. Much higher fluorescence was observed in the non-blocking experiment (left tube, **Fig. 5d**) while negligible fluorescence was detected in the blocking experiment (right tube, **Fig. 5d**). These results have shown that **H4-PEG-PT** had excellent selectivity and targeting ability to osteosarcoma cells.

The following data were added to the revised SI on page 14

Supplementary Fig. 4. The *in vitro* cellular uptake of **H4-PEG-PT** was investigated against 143B cells. Scale bar = 10 μm .

The following sentences were added to the revised SI on page 9

“The cellular uptake of 143B and HepG2 cells treated with H4-PEG-PT

Human bone osteosarcoma 143B cells and human hepatocellular carcinoma HepG2 cells were dispersed in 20 mm confocal dishes and cultured for 12 h at a density of 1×10^5

cells/well. Then, 200 μL of **H4-PEG-PT** (64 μM) was introduced into the confocal dishes. After an incubation time of 7 h, cells were washed two times with PBS, stained with Hoechst staining kit in PBS for 20 min. The tumor cell samples were immediately examined by an inverted Zeiss Axiovert 200 fluorescence microscopy at a 20 \times objective, using a high-pressure mercury lamp as excitation source.”

Besides, we have synthesized another NIR-II probe **CH1055-PEG-PT** to verify the targeting ability due to the targeting peptide PT (*Adv. Healthcare Mater.* **2020**, *9*, 1901224). In our previous paper, **CH1055-PEG-PT** is a known NIR-II fluorophore without mitochondria-targeting ability. **CH1055-PEG-PT** has shown high affinity to osteosarcoma cell lines such as 143B, MG63, SAOS2, but no affinity to HepG2 and EJ cell lines. (*Adv. Healthcare Mater.* **2020**, *9*, 1901224). The oligopeptide PT was crucial to mimic the properties of the natural protein osteocalcin *in vivo* and possessed a high affinity and specificity for osteosarcoma cell lines (*Nanomedicine* 2017 *13*, 111-121). These results from our previous reports and the literature reports suggest the selective targeting ability of NIR-II probe **H4-PEG-PT** against 143B cells due to the targeting peptide PT (PPSHTPT).

(a) NIR-II signals of 143B tumor cells by **CH1055-PEG-PT** with different incubation time. (b) NIR-II signals of 143B cells labelling by **CH1055-PEG-PT** (left) and **CH1055-PEG-PT** + excess PT as a blocking agent (right) under an 808 nm excitation (1000 LP and 100 ms).

(c) NIR-II signals of different cells including 143B, Saos2, MG63, hFOB, EJ, HepG2 by **CH1055-PEG-PT** under an 808 nm excitation (1000 LP and 100 ms). (d) Cellular toxicity of **CH1055-PEG-PT** with different doses (2, 4, 6, 8, 16, 32 μ M) in 143B and hFOB cells. (e) NIR-II fluorescent imaging of **CH1055-PEG-PT** in different media. (f) Compared stability of **CH1055-PEG-PT** and ICG in different media under an 808 nm continuous laser radiation for 30 min (80 mW/cm², 1000 LP).

Reviewers' Comments:

Reviewer #1:

Remarks to the Author:

The authors have adequately addressed the comments raised by this reviewer.

Reviewer #3:

Remarks to the Author:

The reviewer carefully read the manuscript, the previous reviewers' comments and authors' point-to-point reply. The authors provided lots of new data to address the reviewers' concerns and carefully revised manuscript accordingly. The research reported in the study represents the major advancement in the field of NIR-II imaging probe development and highly novel. Thus the reviewer recommend the acceptance of the manuscript for publication in the current form.

Response to reviewer #1

Reviewer #1 (Remarks to the Author):

The authors have adequately addressed the comments raised by this reviewer.

Reply to the reviewer: We very much appreciate Referee #1 for the positive comments.

Response to reviewer #3

Reviewer #3 (Remarks to the Author):

The reviewer carefully read the manuscript, the previous reviewers' comments and authors' point-to-point reply. The authors provided lots of new data to address the reviewers' concerns and carefully revised manuscript accordingly. The research reported in the study represents the major advancement in the field of NIR-II imaging probe development and highly novel. Thus the reviewer recommend the acceptance of the manuscript for publication in the current form.

Reply to the reviewer: We appreciate the reviewer's careful assessment and positive comments of our work, and the recommendation for publishing our paper in Nature Communications.